# Test-time recalibration of conformal predictors under distribution shift based on unlabeled examples

## Abstract

Modern image classifiers are very accurate, but the predictions come without uncertainty estimates. Conformal predictors provide uncertainty estimates by computing a set of classes containing the correct class with a user-specified probability based on the classifier's probability estimates. To provide such sets, conformal predictors often estimate a cutoff threshold for the probability estimates based on a calibration set. Conformal predictors guarantee reliability only when the calibration set is from the same distribution as the test set. Therefore, conformal predictors need to be recalibrated for new distributions. However, in practice, labeled data from new distributions is rarely available, making calibration infeasible. In this work, we consider the problem of predicting the cutoff threshold for a new distribution based on unlabeled examples. While it is impossible in general to guarantee reliability when calibrating based on unlabeled examples, we propose a method that provides excellent uncertainty estimates under natural distribution shifts, and provably works for a specific model of a distribution shift.

## 1 Introduction

Consider a (black-box) image classifier, typically a deep neural network with a softmax layer at the end, that is trained to output probability estimates for $L$ classes given an input feature vector $\mathbf{x} \in \mathbb{R}^d$. Conformal predictors are wrapped around such a classifier and generate a set of classes that contains the correct label with a user-specified probability based on the classifier's probability estimates.

Let $\mathbf{x} \in \mathbb{R}^d$ be a feature vector with associated label $y \in \{1, \ldots, L\}$. We say that a set-valued function $\mathcal{C}$ generates valid prediction sets for the distribution $\mathcal{P}$ if

$$\mathrm{P}_{(\mathbf{x},y)\sim\mathcal{P}} \left[ y \in \mathcal{C}(\mathbf{x}) \right] \geq 1 - \alpha, \tag{1}$$

where $1 - \alpha$ is the desired coverage level. Conformal predictors generate valid sets $\mathcal{C}$ for the distribution $\mathcal{P}$ by utilizing a calibration set consisting of labeled examples $\{(\mathbf{x}_1, y_1), \ldots, (\mathbf{x}_n, y_n)\}$. An important caveat of conformal predictors is that the examples from the calibration set are drawn from the same distribution as the test dataset.

This assumption is difficult to satisfy in applications and potentially limits the applicability of conformal prediction methods in practice. In fact, in practice one usually expects a distribution shift between the calibration set and the examples at inference (or the test set), in which case the coverage guarantees provided by conformal prediction methods are void. For example, the new ImageNetV2 test set was created in the same way as the original ImageNet test sets, yet Recht et al. (2019) found a notable drop in classification accuracy for all classifiers considered.

Ideally, a conformal predictor is recalibrated on a distribution before testing, otherwise the coverage guarantees are not valid (Cauchois et al., 2020). However, in real-world applications, where distribution shifts are ubiquitous, labeled data from new distributions is scarce or non-existent.

We therefore consider the problem of recalibrating a conformal predictor only based on unlabeled data from the new domain. This is an ill-posed problem: it is in general impossible to calibrate a conformal predictor

based on unlabeled data. Yet, we propose a simple calibration method that gives excellent performance for a variety of natural distribution shifts.

**Organization and contributions.** We start with concrete examples on how conformal predictors yield miscalibrated uncertainty estimates under natural distribution shifts. We next propose a simple recalibration method that only uses unlabeled examples from the target distribution. We show that our method correctly recalibrates a popular conformal predictor (Sadinle et al., 2019) on a theoretical toy model. We provide empirical results for various natural distribution shifts of ImageNet showing that recalibrating conformal predictors using our proposed method significantly reduces the performance gap. In certain cases, it even achieves near oracle-level coverage.

**Related work.** Several works have considered the robustness of conformal prediction to distribution shift (Tibshirani et al., 2019; Gibbs & Candes, 2021; Park et al., 2022; Barber et al., 2023; Prinster et al., 2022; 2023; Gibbs & Candès, 2023; Fannjiang et al., 2022). Gibbs & Candes (2021); Gibbs & Candès (2023) consider a setting where the distribution varies over time and propose an adaptive conformal prediction method to guarantee asymptotic and local coverage. Similarly, Barber et al. (2023) propose a weighted conformal prediction method to provably generalize to the case where distribution changes over time. On the other hand, Prinster et al. (2022; 2023) propose a weighted uncertainty quantification based on the jackknife+ method rather than the typical conformal prediction methods that we consider in this paper.

Particularly of interest, Tibshirani et al. (2019) and Park et al. (2022) propose methods that assume a covariate shift and calibrate based on estimating the amount of covariate shift, we compare to those later in Section 5.2. Podkopaev & Ramdas (2021) studies the related, but discrete setting of label shifts between the source and target domains and proposes a method that is more robust under the label shift setting. In contrast, we focus on complex image datasets for which covariate shift is not well defined and label shift not broadly relevant.

We are not aware of other works studying calibration of conformal predictors under distribution shift based on unlabeled examples. However, prior works propose to make conformal predictors robust to various distribution shifts from the source distribution of the calibration set (Cauchois et al., 2020; Gendler et al., 2022), via calibrating the conformal predictor to achieve a desired coverage in the worse case scenario of the considered distribution shifts. Cauchois et al. (2020) considers covariate shifts and calibrates the conformal predictor to achieve coverage for the worst-case distribution within the $f$-divergence ball of the source distribution. Gendler et al. (2022) considers adversarial perturbations as distribution shifts and calibrates a conformal predictor to achieve coverage for the worst-case distribution obtained through $\ell_2$-norm bounded adversarial noise.

While making the conformal predictor robust to a range of worst-case distributions at calibration time allows maintaining coverage under the worst-case distributions, these approaches have two shortcomings: First, natural distribution shifts are difficult to capture mathematically, and models like covariate-shifts or adversarial perturbations do not seem to model natural distribution shifts (such as that from ImageNet to ImageNetV2) accurately. Second, calibrating for a worst-case scenario results in an overly conservative conformal predictor that tends to yield much higher coverage than desired for test distributions that correspond to a less severe shift from the source, which comes at the cost of reduced efficiency (i.e., larger set size, or larger confidence interval length). In contrast, our method does not compromise the efficiency of the conformal predictor on easier distributions as we recalibrate the conformal predictor for any new dataset.

A related problem is to predict the accuracy of a classifier on new distributions from unlabeled data sampled from a new distribution (Deng & Zheng, 2021; Chen et al., 2021; Jiang et al., 2022; Deng et al., 2021; Guillory et al., 2021; Garg et al., 2022). In particular, Garg et al. (2022) proposed a simple method that achieves state-of-the-art performance in predicting classifier accuracy across a range of distributions. However, the calibration problem we consider is fundamentally different than estimating the accuracy of a classifier. While predicting the accuracy of the classifier would allow making informed decisions on whether to use the classifier for a new distribution, it doesn't provide a solution for recalibration.

## 2 Background on conformal prediction

Consider a black-box classifier with input feature vector $\mathbf{x} \in \mathbb{R}^d$ that outputs a probability estimate $\pi_\ell(\mathbf{x}) \in [0, 1]$ for each class $\ell = 1, \ldots, L$. Typically, the classifier is a neural network trained on some distribution, and the probability estimates are the softmax outputs. We denote the order statistics of the probability estimates by $\pi_{(1)}(\mathbf{x}) \geq \pi_{(2)}(\mathbf{x}) \geq \ldots \geq \pi_{(L)}(\mathbf{x})$.

Many conformal predictors use a calibration set $\mathcal{D}_{\text{cal}}^{\mathcal{P}} = \{(\mathbf{x}_i, y_i)\}_{i=1}^n$ to find a cutoff threshold (Sadinle et al., 2019; Romano et al., 2020; Angelopoulos et al., 2020; Bates et al., 2021) that achieves the desired empirical coverage on this set. Here, the superscript $\mathcal{P}$ denotes the distribution from which the examples in the calibration set are sampled from. Given a set-valued function $\mathcal{C}(\mathbf{x}, u, \tau) \subset \{1, \ldots, L\}$ containing the set of predicted classes by the conformal predictor, such conformal predictors compute the threshold parameter $\tau$ as

$$\tau^* = \inf \left\{ \tau : |\{i : y_i \in \mathcal{C}(\mathbf{x}_i, u_i, \tau)\}| \geq (1 - \alpha)(n + 1) \right\}, \tag{2}$$

where $u_i$ is added randomization to smoothen the cardinality term, chosen independently and uniformly from the interval $[0, 1]$, see Vovk et al. (2005) on smoothed conformal predictors. Finally, the '+1' term in the $(n + 1)$ term is a bias correction for the finite size of the calibration set.

This conformal calibration procedure achieves distributional coverage as defined in the expression (1), for any set valued function $\mathcal{C}(\mathbf{x}, u, \tau)$ satisfying the nesting property $\mathcal{C}(\mathbf{x}, u, \tau_1) \subseteq \mathcal{C}(\mathbf{x}, u, \tau_2)$ for $\tau_1 < \tau_2$, see (Angelopoulos et al., 2020, Thm. 1).

In this paper, we primarily focus on the popular conformal predictors *Thresholded Prediction Sets* (TPS) (Sadinle et al., 2019) and *Adaptive Prediction Sets* (APS) (Romano et al., 2020). The set generating functions of the two conformal predictors are

$$\mathcal{C}^{\text{TPS}}(\mathbf{x}, \tau) = \{\ell = 1, \ldots, L : \pi_\ell(\mathbf{x}) \geq 1 - \tau\}, \tag{3}$$

$$\mathcal{C}^{\text{APS}}(\mathbf{x}, u, \tau) = \{\ell = 1, \ldots, L : \sum_{j=1}^{\ell-1} \pi_{(j)}(\mathbf{x}) + u \cdot \pi_{(\ell)}(\mathbf{x}) \leq \tau\}, \tag{4}$$

with $u \sim U(0, 1)$ for smoothing. The set generating function of TPS doesn't require smoothing since each softmax score is independently thresholded and therefore there are no discrete jumps.

Computing the threshold $\tau$ through conformal calibration (2) requires a labeled calibration set from distribution $\mathcal{P}$. We therefore add a superscript to the threshold to designate which distribution the calibration set was sampled from; for example $\tau^{\mathcal{P}}$ indicates that the calibration set was sampled from the distribution $\mathcal{P}$. The prediction set function $\mathcal{C}^{\text{TPS}}$ for TPS and $\mathcal{C}^{\text{APS}}$ for APS both satisfy the nesting property. Therefore, TPS and APS calibrated on a calibration set $\mathcal{D}_{\text{cal}}^{\mathcal{P}}$ by computing the threshold in the expression (2) is guaranteed to achieve coverage on the distribution $\mathcal{P}$. However, coverage is only guaranteed if the test distribution $\mathcal{Q}$ is the same as the calibration distribution $\mathcal{P}$.

## 3 Failures under distribution shifts and problem statement

Often we are most interested in quantifying uncertainty with conformal prediction when we apply a classifier to new data that might come from a slightly different distribution than the distribution we calibrated on. Yet, conformal predictors only provide coverage guarantees for data coming from the same distribution as the calibration set, and the coverage guarantees often fail even under slight distribution shifts. For example, our experiments (see Figure 3) show that APS calibrated on ImageNet-Val to yield $1 - \alpha = 0.9$ coverage only achieves a coverage of 0.64 on the ImageNet-Sketch dataset, which consists of sketches of the ImageNet-Val images and hence constitutes a distribution shift (Wang et al., 2019).

Different conformal predictors typically have different coverage gaps under the same distribution shift. More efficient conformal predictors (i.e., those that produce smaller prediction sets) tend to have a larger coverage gap under a distribution shift. For example, both TPS and RAPS (a generalization of APS proposed by Angelopoulos et al. (2020)) yield smaller confidence sets, but only achieve a coverage of 0.38 vs. 0.64 for APS on the ImageNet-Sketch distribution shift discussed above.

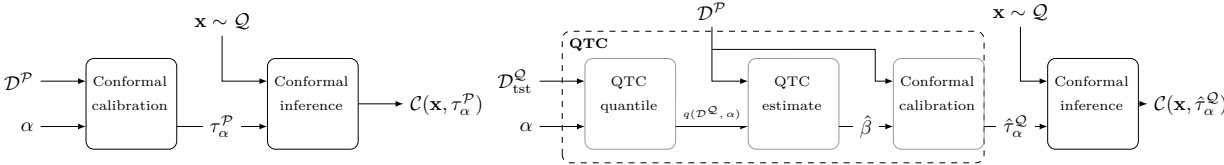

Figure 1: **Left**: Vanilla conformal prediction. **Right**: QTC recalibration. QTC encapsulates the conformal calibration process to recalibrate the conformal predictor for each new distribution without altering the underlying set generating function. $\mathcal{D}^{\mathcal{Q}}_{\text{tst}}$ is the unlabeled test set and $\mathcal{D}^{\mathcal{P}}$ is the labeled training/calibration set. QTC finds a threshold on the scores of the model on the unlabeled samples and predicts the coverage level by utilizing how the distribution of the scores changes across test distribution with respect to this threshold.

Even under more subtle distribution shifts such as subpopulation shifts (Santurkar et al., 2021), the achieved coverage can drop significantly. For example, APS calibrated to yield $1 - \alpha = 0.9$ coverage on the source distribution of the Living-17 BREEDS dataset only achieves a coverage of 0.68 on the target distribution. The source and target distributions contain images of exclusively different breeds of animals while the animals' species is shared as the label (Santurkar et al., 2021).

**Problem statement.** Our goal is to recalibrate a conformal predictor on a new distribution $\mathcal{Q}$ based on unlabeled data. Given an unlabeled dataset $\mathcal{D}^{\mathcal{Q}}_{\text{tst}} = \{\mathbf{x}_1, \dots, \mathbf{x}_n\}$ sampled from the target distribution $\mathcal{Q}$, our goal is to provide an accurate estimate $\hat{\tau}^{\mathcal{Q}}$ for the threshold $\tau^{\mathcal{Q}}$. Recall that the threshold $\tau^{\mathcal{Q}}$ is so that the conformal predictor with set function $\mathcal{C}(\mathbf{x}, u, \tau^{\mathcal{Q}})$ achieves the desired coverage of $1 - \alpha$ on the target distribution $\mathcal{Q}$. Thus, in other words, our goal is to estimate a threshold $\hat{\tau}^{\mathcal{Q}}$ so that the set $\mathcal{C}(\mathbf{x}, u, \hat{\tau}^{\mathcal{Q}})$ achieves close to the desired coverage of $1 - \alpha$ on the target distribution, based on the unlabeled dataset only.

In general, it is impossible to guarantee coverage since conformal prediction relies on exchangeability assumptions which can not be guaranteed in practice for new datasets (Vovk et al., 2005; Romano et al., 2020; Angelopoulos et al., 2020; Cauchois et al., 2020; Bates et al., 2021). However, we will see that we can consistently estimate the threshold $\tau^{\mathcal{Q}}$ for a variety of natural distribution shifts.

We refer to the difference between the target coverage of $1 - \alpha$ and the actual coverage achieved on a given distribution without any recalibration efforts as the *coverage gap*. We assess how effective a recalibration method is based on the reduction of the coverage gap after recalibration.

## 4 Methods

In this section we introduce our calibration method, termed Quantile Thresholded Confidence (QTC), along with baseline methods we consider in our experiments.

### 4.1 Quantile thresholded confidence

Consider a conformal predictor with threshold $\tau^{\mathcal{P}}_{\alpha}$ calibrated so that the conformal predictor achieves coverage $1 - \alpha$ on the source distribution $\mathcal{P}$. On a different distribution $\mathcal{Q}$ the coverage of the conformal predictor is off. But there is a value $\beta$ such that, if we calibrate the conformal predictor on the *source distribution* using the value $\beta$ instead of $\alpha$, it achieves $1 - \alpha$ coverage on the *target distribution*, i.e., the corresponding thresholds obey $\tau^{\mathcal{P}}_{\beta} = \tau^{\mathcal{Q}}_{\alpha}$.

Our method first estimates the value $\beta$ based on unlabeled examples. From the estimate $\hat{\beta}$, we estimate $\tau^{\mathcal{Q}}_{\alpha}$ based on computing the threshold $\tau^{\mathcal{P}}_{\hat{\beta}}$ by calibrating the conformal predictor on the source calibration set using $\hat{\beta}$. This yields a threshold close to the desired one, i.e., $\tau^{\mathcal{P}}_{\hat{\beta}} \approx \tau^{\mathcal{Q}}_{\alpha}$.

**Step 1, estimation of $\beta$:** We are given a labeled source dataset $\mathcal{D}_{\mathrm{cal}}^{\mathcal{P}}$ and an unlabeled target dataset $\mathcal{D}_{\mathrm{tst}}^{\mathcal{Q}}$. Our estimate of $\beta$ relies on the quantile function

$$q(\mathcal{D}, c) = \inf \left\{ p \colon \frac{1}{|\mathcal{D}|} \sum_{\mathbf{x} \in \mathcal{D}} \mathbb{1}_{\{s(\pi(\mathbf{x})) < p\}} \geq c \right\}. \tag{5}$$

The quantile function depends on the classifier's predictions through a score function $s(\pi(\mathbf{x})) = \max_\ell \pi_\ell(\mathbf{x})$, which we take as the largest softmax score of the classifier's predictions. Here, $\mathcal{D}$ is a set of unlabeled examples and $c \in [0, 1]$ is a scalar. Our method first identifies a threshold based on the unlabeled target dataset $\mathcal{D}_{\mathrm{tst}}^{\mathcal{Q}}$ for a desired coverage level $\alpha$ in expression (5) by computing $q(\mathcal{D}_{\mathrm{tst}}^{\mathcal{Q}}, \alpha)$. Since this process is identical to finding the $(\alpha)^{th}$ quantile of the scores on the dataset, we dub the method Quantile Thresholded Confidence (QTC). QTC estimates $\beta$ as

$$\beta_{\mathrm{QTC}} = \min(\beta_{\mathrm{QTC-T}}, \beta_{\mathrm{QTC-S}}), \tag{6}$$

where the QTC-Target and QTC-Source estimates are

$$\beta_{\mathrm{QTC-T}}(\mathcal{D}_{\mathrm{tst}}^{\mathcal{Q}}) = \frac{1}{|\mathcal{D}_{\mathrm{cal}}^{\mathcal{P}}|} \sum_{\mathbf{x} \in \mathcal{D}_{\mathrm{cal}}^{\mathcal{P}}} \mathbb{1}_{\left\{ s(\pi(\mathbf{x})) < q(\mathcal{D}_{\mathrm{tst}}^{\mathcal{Q}}, \alpha) \right\}} \tag{7}$$

$$\beta_{\mathrm{QTC-S}}(\mathcal{D}_{\mathrm{tst}}^{\mathcal{Q}}) = 1 - \frac{1}{|\mathcal{D}_{\mathrm{tst}}^{\mathcal{Q}}|} \sum_{\mathbf{x} \in \mathcal{D}_{\mathrm{tst}}^{\mathcal{Q}}} \mathbb{1}_{\left\{ s(\pi(\mathbf{x})) < q(\mathcal{D}_{\mathrm{cal}}^{\mathcal{P}}, 1 - \alpha) \right\}}. \tag{8}$$

We consider two estimates for $\beta$, and aggregate them to a single value by taking the minimum of the two. This yields best performance, as demonstrated by studying the three versions of QTC, corresponding to the three estimates (6), (7), and (8).

The reasons for having two estimates and aggregating them is as follows. DNNs have a tendency to be over-confident in their predictions (Guo et al., 2017). If the distribution of the softmax scores over the dataset is not sufficiently smooth in the lower-confidence regime, the QTC-T estimate might be inaccurate. In this higher-confidence regime QTC-S provides a better estimate. The minimum of the two provides a good estimate in the high and low confidence regions.

The motivation behind QTC is that we essentially map the quantile function conformal prediction uses, which relies on the labels, to the quantile function of QTC, which does not require labels. While this mapping is not guaranteed to be preserved under distribution shift, we have observed that it works very well in practice and provably works in the theoretical setting that we consider.

If there is no distribution shift between the source and target, QTC would recover the original $\alpha$. That is, both the QTC-T and QTC-S estimates of the $\beta$ would be asymptotically equal to $\alpha$ and $1 - \alpha$ respectively. To see this more clearly, note that we can insert the definition of $q$ in (5) in the RHS of the equations (7), (8). As $n \to \infty$, the sums over the datasets converge to the expectations which are equal when no distribution shift is present.

**Step 2, estimation of the threshold $\tau_\alpha^{\mathcal{Q}}$ based on $\beta$:** QTC predicts the conformal threshold $\tau_\alpha^{\mathcal{Q}}$ by conformal calibration with target value $\beta_{\mathrm{QTC}}$. Specifically, we calibrate the conformal predictor on the dataset $\mathcal{D}_{\mathrm{cal}}^{\mathcal{P}}$ as

$$\tau_{\mathrm{QTC}} = \inf \left\{ \tau \colon |\{i \colon y_i \in \mathcal{C}(\mathbf{x}_i, u_i, \tau)\}| \geq (1 - \beta_{\mathrm{QTC}})(|\mathcal{D}_{\mathrm{cal}}^{\mathcal{P}}| + 1) \right\}, \tag{9}$$

which yields the estimate $\tau_{\mathrm{QTC}}$ for $\tau_\alpha^{\mathcal{Q}}$. QTC is illustrated in Figure 1.

QTC is inspired by a method for predicting a classifier's accuracy from Garg et al. (2022). Garg et al. (2022)'s method finds a threshold on the scores matching the accuracy of a classifier on the dataset and predicts the accuracy on other datasets. Contrary, we predict the threshold of a conformal predictor, and our method is based on predicting an auxiliary parameter $\beta$ instead of a threshold directly.

### 4.2 Baseline methods

We consider regression-based methods as baselines. Regression-based methods have been used for predicting classification accuracy, assuming a correlation between the classification accuracy and a feature (e.g., average confidence) across different distributions (Deng et al., 2021; Deng & Zheng, 2021; Guillory et al., 2021). We consider regression-based methods as baselines for predicting the conformal threshold on a target distribution that would achieve $1 - \alpha$ coverage. We train the regression-based methods on a dataset consisting of synthetically generated distributions given a source distribution (e.g. ImageNet-C from ImageNet) with the goal of predicting the conformal threshold for a test dataset sampled from a natural distribution.

Let $\phi_\pi(\mathcal{D}) \colon \mathbb{R}^L \to \mathbb{R}^d$ be the feature extractor part of a neural network that maps the softmax scores of the classifier to the features for a given dataset $\mathcal{D}$. A simple example is the one-dimensional feature $(d = 1)$ extracted by computing the average confidence of a given classifier across the examples of a given dataset.

We fit a regression function $f_\theta$ parameterized by different feature extractors $\phi_\pi$ by minimizing the mean squared error between the output and the calibrated threshold $\tau$ across the distributions as

$$\hat{\theta} = \arg\min_\theta \sum_j (f_\theta(\phi_\pi(\mathcal{D}_j)) - \tau^{\mathcal{P}_j})^2. \tag{10}$$

We consider the following choices for the feature extractor $\phi_\pi$ (see App A.1 for details):

- *Average confidence regression (ACR)*: The average confidence of the classifier across the entire dataset.

- *Difference of confidence regression (DCR)* (Guillory et al., 2021): The average confidence of the classifier across the entire dataset offset by the average confidence on the source dataset. Prediction is also for the offset target $\tau - \tau^{\mathcal{P}}$. DCR performs better than ACR for predicting a classifier's accuracy (Guillory et al., 2021).

- *Confidence histogram-density regression (CHR)*: Normalized histogram density of the classifier confidence across the dataset, where the feature dimension is controlled by a hyperparameter that determines the number of histogram bins in the probability range $[0, 1]$. Neural networks tend to be overconfident in their prediction which heavily skews the histogram densities to the last bin. We also therefore consider a variant of CHR, *dubbed CHR-*, where we drop the last bin of the histogram as a feature.

- *Predicted class-wise average confidence regression (PCR)*: Class-wise (by predicted class) average confidence of the classifier across the samples.

## 5 Experiments

We study the performance of QTC on natural distribution shifts and on an artifical covariate shift.

### 5.1 Natural distribution shifts

We consider the following choices for the source distribution $\mathcal{P}$ and associated natural distribution shifts:

**ImageNet (Deng et al., 2009) distribution shifts:** In our ImageNet experiments, ImageNet is the source distribution $\mathcal{P}$ and the following natural distribution shifts are the target distributions $\mathcal{Q}$:

- **ImageNetV2** (Recht et al., 2019) was constructed by following the same procedure as for constructing and labeling the original ImageNet dataset. However, all standard models perform significantly worse on ImageNetV2 relative to the original ImageNet test set.

- **ImageNet-Sketch** (Wang et al., 2019) contains sketch-like images of the objects in the original ImageNet, but otherwise matches the original categories and scales.

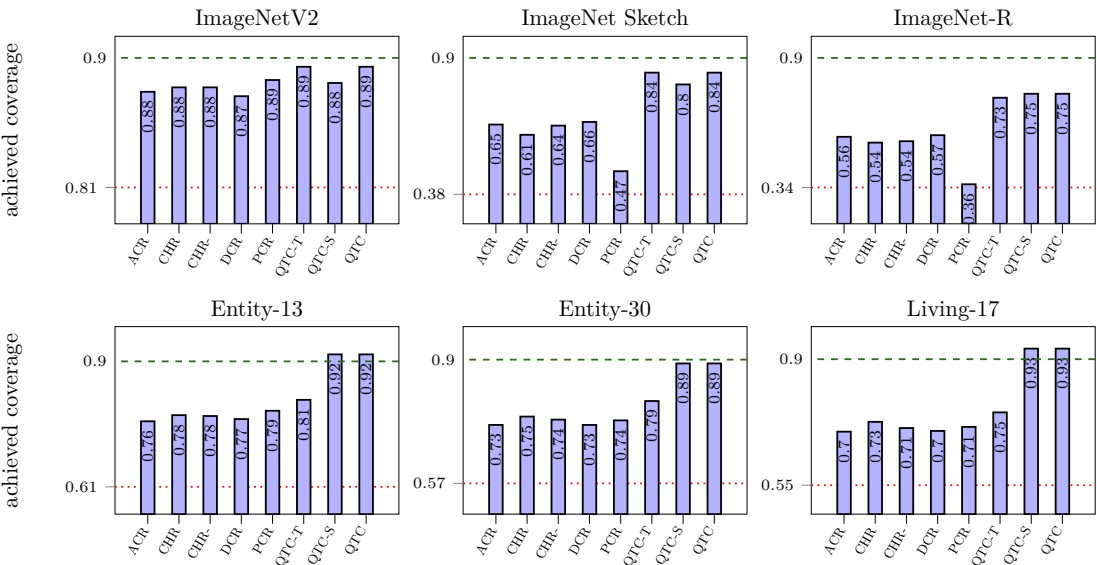

Figure 2: Coverage obtained by TPS for a desired coverage of $1 - \alpha = 0.9$ on the target distribution $\mathcal{Q}$ after recalibration using the unlabeled samples from $\mathcal{Q}$ for various recalibration methods. The dotted line is the coverage without recalibration, and the dashed line is the target coverage $1 - \alpha = 0.9$. The figure shows that QTC-T and QTC-S almost fully close the coverage gap across ImageNet and BREEDS test distribution shifts, corresponding to varying severities.

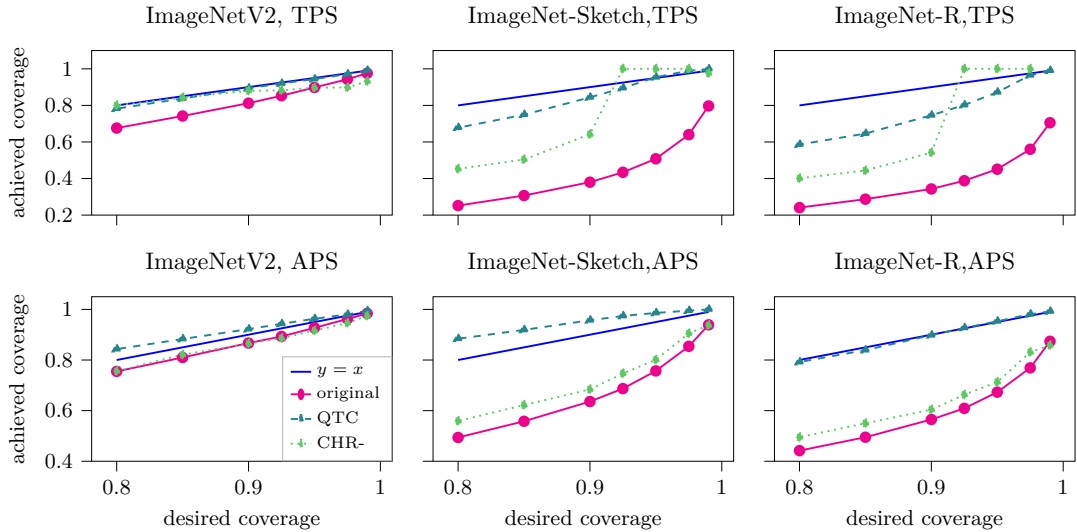

Figure 3: Coverage obtained by TPS and APS on the target distribution $\mathcal{Q}$ as a function of the desired coverage (i.e., $1 - \alpha$) after recalibration with the respective prediction method. For regression methods, only the best performing method, CHR-, is shown. QTC significantly closes the coverage gap across the range of $1 - \alpha$, while CHR- yields inconsistent or insufficient performance improvements.

- **ImageNet-R** (Hendrycks et al., 2021) contains artwork images of the ImageNet class objects found in the web. ImageNet-R only contains images for a 200-class subset of the original ImageNet. We don't limit our experiments to this subset but instead consider the adverse setting of calibrating on all 1000 classes since our main goal is to provide an end-to-end solution for recalibration of the conformal predictors and we are interested in how well our method performs against possible adversaries such as dataset imbalance that can be encountered in practice.

**BREEDS (Santurkar et al., 2021) distribution shifts:** The BREEDS datasets feature *sub-population shifts* from the training set to test. The BREEDS datasets were constructed using the existing ImageNet images, but with different classes. BREEDS utilizes the hierarchical WordNet structure of the classes to choose a parent class that makes the original ImageNet classes the leaves. For example, in the BREEDS Living-17 dataset, one of the classes is *domestic cat*. This is a parent class of several ImageNet classes, which are *tiger cat, Egyptian cat, Persian cat and Siamese cat*. BREEDS induces a subpopulation shift from the source distribution to the target by assigning these leaf classes to either the source or target. For example, the images in the source dataset of Living-17 under the *domestic cat* class are that of either *tiger cats* or *Egyptian cats*, whereas in the target are that of either *Persian cats* or *Siamese cats*. Therefore, despite having the same label (*domestic cat*), the source and target distributions semantically differ due to the differences between the breeds, which induces a subpopulation shift.

We consider three BREEDS datasets: Entity-13, Entity-30 and Living-17, which are named using the convention *theme/object type–#classes*.

**Experimental procedure.** For the ImageNet experiments we use a ResNet-50 and DenseNet-121 pretrained on the ImageNet training set. For the BREEDS experiments, we train a ResNet-18 model from scratch for the BREEDS datasets. In both cases, the classifiers only see examples from the source distribution.

For all experiments, we first calibrate the conformal predictor on the source distribution $\mathcal{P}$ to find the cutoff threshold $\tau^{\mathcal{P}}$. For QTC and variants, we find the threshold $q$ using the expression (5). For the regression methods, we use the ImageNet-C dataset (Hendrycks & Dietterich, 2019) as the source of synthetic distributions, find the cutoff threshold $\tau$ for each of the distributions, and fit a regressor by minimizing the loss (10). For the regression function we use a 4-layer MLP with ReLU activations. ImageNet-C consists of 90 different distributions obtained by synthetically perturbing the images of ImageNet-Val for 18 different types of perturbations at 5 different levels of severity, resulting in 90 distinct distributions.

**Recalibration experiments for a fixed target coverage.** We first evaluate the recalibration methods for a fixed target coverage of $1 - \alpha = 0.9$. The results in Figure 2 for recalibrating TPS show that QTC reduces the coverage gap much more than regression methods, and even closes it in some cases.

We also display QTC-T and QTC-S as ablation studies. Here it can be seen that sometimes QTC-T and sometimes QTC-S performs best, which is why combining them is necessary. The different performance of QTC-T and QTC-S can be attributed to the difference of the type of shifts (e.g. semantic vs. subpopulation) between ImageNet and BREEDS. Note that QTC-T operates on the regime of samples with lower confidence whereas QTC-S on the higher confidence regime. Therefore, QTC-T may perform subpar compared to QTC-S for datasets consisting of fewer, more distinct classes like BREEDS, for which a well-trained classifier tends to assign high confidence to its predictions.

**Recalibration experiments for different target coverage levels.** The coverage gap (i.e., the difference of achieved coverage and targeted coverage) varies across the desired coverage level $1 - \alpha$. We therefore next evaluate the performance as a function of the desired coverage level.

Figure 3 shows the coverage obtained after recalibration with TPS and APS for different values of $1 - \alpha$ for the natural distribution shifts from ImageNet. QTC closes the coverage gap significantly for all choices of $1 - \alpha$, whereas the best performing regression-based baseline method, CHR-, fails to significantly improve the coverage gap consistently across all choices of $1 - \alpha$.

## 5.2 Comparison to covariate shift based methods

QTC does not require labeled data from the target distribution at training or inference time. Existing methods that aim to measure the amount of covariate shift based on unlabeled examples also improve the robustness of conformal prediction, but rely on labeled examples from the target domain (Tibshirani et al., 2019; Park et al., 2022). Here, we compare the performance of QTC to that of covariate shift based methods and show that QTC outperforms the state-of-the-art when labeled data is not available during training, and performs only marginally worse if labeled data is available.

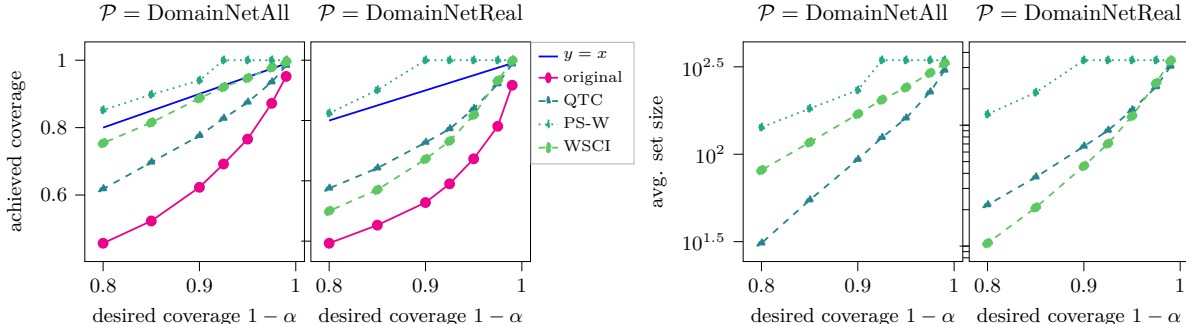

Figure 4: Coverage (**left**) and the average set size (**right**) obtained by TPS on the target $\mathcal{Q} =$ DomainNet-Infograph for various settings of $(1 - \alpha)$. For the setting where all domains are available for the discriminator (**left**), WSCI closes the coverage gap while QTC considerably improves it; whereas when only DomainNet-Real is available, QTC slightly outperforms. In both settings, PS-W fails by constructing uninformatively large confidence sets for the range $1 - \alpha > 0.9$.

Under a covariate shift, the conditional distribution of the label $y$ given the feature vector $\mathbf{x}$ is fixed but the marginal distribution of the feature vectors differ:

$$\text{source:}(\mathbf{x}, y) \sim \mathcal{P} = p_{\mathcal{P}}(\mathbf{x}) \times p(y|\mathbf{x}), \qquad \text{target:}(\mathbf{x}, y) \sim \mathcal{Q} = p_{\mathcal{Q}}(\mathbf{x}) \times p(y|\mathbf{x}),$$

where $p_{\mathcal{P}}(\mathbf{x})$ and $p_{\mathcal{Q}}(\mathbf{x})$ are the marginal PDFs of the features $\mathbf{x}$, and $p(y|\mathbf{x})$ is the conditional PDF of the label $y$.

In order to account for a covariate shift, Tibshirani et al. (2019); Park et al. (2022) utilize an approach called *weighted conformal calibration*. Weighted conformal calibration uses the likelihood ratio of the covariate distributions, i.e., the importance weights $w(\mathbf{x}) = p_{\mathcal{Q}}(\mathbf{x})/p_{\mathcal{P}}(\mathbf{x})$ to weigh the scores used for the set generating function of the conformal predictor for each sample $(\mathbf{x}, y) \in \mathcal{D}_{\text{cal}}^{\mathcal{P}}$. A conformal predictor calibrated on a source calibration set with the true importance weights for a target distribution is guaranteed to achieve the desired coverage on the target, see Tibshirani et al. (2019, Cor. 1). In practice, the importance weights are not known and are therefore estimated heuristically.

Covariate shifts is not well defined for complex tasks such as image classification. We therefore follow the experimental setup of Park et al. (2022) and consider a backbone ResNet-101 classifier trained using unsupervised domain adaptation based on training sets sampled from both the source and target distribution as well as an auxillary classifier (discriminator) $g$ that yields probability estimates of membership between the two for a given sample. For the *weighted split conformal inference* (WSCI) method of Tibshirani et al. (2019), we estimate the importance weights using this discriminator $g$ and for the PAC prediction sets method of Park et al. (2022) based on rejection sampling (PS-W), using histogram density estimation over the probability estimates. We use TPS as the conformal predictor.

We consider the DomainNet distribution shift problem (Peng et al., 2019) and choose *DomainNet-Infograph* as the target distribution since the coverage gap is insignificant for the others (see Park et al. (2022, Table 1)). We consider two scenarios, for both of which all six DomainNet domains, i.e. *DomainNet-Sketch, DomainNet-Clipart, DomainNet-Painting, DomainNet-Quickdraw, DomainNet-Real, and DomainNet-Infograph*, are available during training. In the first scenario all domains are also available at inference, whereas in the second scenario, analogous to the ImageNet setup, we only have access to the examples from *DomainNet-Real* (source) and *DomainNet-Infograph* (target).

The results in Figure 4 show that when the source includes all the domains, WSCI outperforms other methods. However, when only DomainNet-Real is available for the source at calibration time, QTC slightly outperforms WSCI. In both settings, PS-W fails if $\alpha$ is chosen such that $1 - \alpha > 0.9$, by constructing uninformatively large confidence sets that tend to contain all possible labels. On the other hand, QTC and WSCI tend to construct similarly sized confidence sets consistently across the range of $1 - \alpha$. Note that while QTC considerably closes the coverage gap in both setups, QTC-S fails to improve the coverage gap. This might be due to the fact that

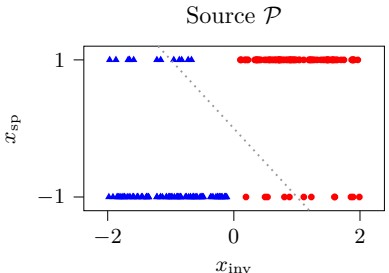 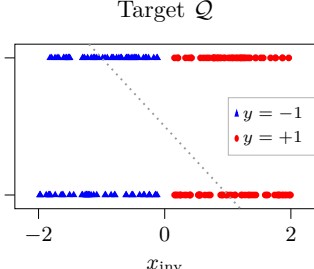

Figure 5: Example source and target distributions $\mathcal{P}$ and $\mathcal{Q}$ for the binary classification model, and a classifier with $w_{\text{inv}}, w_{\text{sp}} = 1$. The decision boundary is shown with a faded dotted line. The correlation between the feature $x_{\text{sp}}$ and the label $y$ is higher for the source than target ($p^{\mathcal{P}} > p^{\mathcal{Q}}$).

ResNet-101 trained with domain adaptation tends to yield very high confidence across all examples. While a separate discriminator that uses the representations of the ResNet-101 before the fully-connected linear layer is utilized for the covariate shift based methods, this is not the case for QTC and its variants. Therefore, the threshold found by QTC-S tends to be very close or even equal to 1.0, hindering the performance.

## 6 Theoretical results

We consider a simple binary classification distribution shift model from Nagarajan et al. (2021); Garg et al. (2022), and adapt the analysis from Garg et al. (2022) to show that recalibrating provably succeeds within this model. Specifically, we show that the conformal predictor TPS with QTC-T yields the desired coverage of $1 - \alpha$ on the target distribution based on unlabeled examples.

The distribution shift model from Nagarajan et al. (2021) is as follows. Consider a binary classification problem with response $y \in \{-1, 1\}$ and with two features $\mathbf{x} = [x_{\text{inv}}, x_{\text{sp}}] \in \mathbb{R}^2$, an invariant one and a spuriously correlated one. The source and target distributions $\mathcal{P}$ and $\mathcal{Q}$ over the feature vector and label are defined as follows. The label $y$ is uniformly distributed over $\{-1, 1\}$. The invariant fully-predictive feature $x_{\text{inv}}$ is uniformly distributed in an interval determined by the constants $c > \gamma \geq 0$, with the interval being conditional on $y$:

$$x_{\text{inv}}|y \sim \begin{cases} U\left[\gamma, c\right] & \text{if } y = 1 \\ U\left[-c, -\gamma\right] & \text{if } y = -1 \end{cases}. \tag{11}$$

The spurious feature $x_{\text{sp}}$ is correlated with the response $y$ such that $\mathrm{P}_{(\mathbf{x},y)\sim\mathcal{P}}\left[x_{\text{sp}} \cdot y > 0\right] = p^{\mathcal{P}}$, where $p^{\mathcal{P}} \in (0.5, 1.0)$ for some joint distribution $\mathcal{P}$. A distribution shift is modeled by simulating target data with different degrees of spurious correlation such that $\mathrm{P}_{(\mathbf{x},y)\sim\mathcal{Q}}\left[x_{\text{sp}} \cdot y > 0\right] = p^{\mathcal{Q}}$, where $p^{\mathcal{Q}} \in [0, 1]$. There is a distribution shift from source to target when $p^{\mathcal{P}} \neq p^{\mathcal{Q}}$. Two example distributions $\mathcal{P}$ and $\mathcal{Q}$ are illustrated in Figure 5.

We consider a logistic regression classifier that predicts class probability estimates for the classes $y = -1$ and $y = 1$ as $f(\mathbf{x}) = \left[\frac{1}{1+e^{\mathbf{w}^T\mathbf{x}}}, \frac{e^{\mathbf{w}^T\mathbf{x}}}{1+e^{\mathbf{w}^T\mathbf{x}}}\right]$, where $\mathbf{w} = [w_{\text{inv}}, w_{\text{sp}}] \in \mathbb{R}^2$. The classifier with $w_{\text{inv}} > 0$ and $w_{\text{sp}} = 0$ minimizes the misclassification error across all choices of distributions $\mathcal{P}$ and $\mathcal{Q}$ (i.e., across all choices of $p$). However, a classifier learned by minimizing the empirical logistic loss via gradient descent depends on both the invariant feature $x_{\text{inv}}$ and the spuriously-correlated feature $x_{\text{sp}}$, i.e., $w_{\text{sp}} \neq 0$ due to the geometric skews on the finite data and statistical skews of the optimization with finite gradient descent steps (Nagarajan et al., 2021).

For the logistic regression classifier TPS recalibrated with QTC-T provably suceeds:

**Theorem 6.1** (Informal). *Consider the logistic regression classifier for the binary classification problem described above with $w_{\text{inv}} > 0, w_{\text{sp}} \neq 0$, let $n$ be the number of samples for the source and target datasets and $\alpha \in (0, \epsilon)$ be a user-defined value, where $\epsilon$ is the error rate of the classifier on the source. The coverage achieved on the target by recalibrating TPS on the source with the QTC estimate obtained in (7) by finding the QTC threshold on the target as in (5) converges to $1 - \alpha$ as $n \to \infty$ with high probability.*

Regarding the assumption on $\alpha$: A value of $\alpha$ that is larger than the error rate of the classifier does make sense as it would result in empty confidence sets for a portion of the examples in the dataset.

In order to understand the intuition behind Theorem 6.1, we first explain how the coverage is off under a distribution shift in this model. Consider a classifier that depends positively on the spurious feature (i.e., $w_{\mathrm{sp}} > 0$). When the spurious correlation is decreased from the source distribution to the target, the error rate of the classifier increases. TPS calibrated on the source samples finds a threshold $\tau$ such that the prediction sets yield $1 - \alpha$ coverage on the source dataset as $n \to \infty$. In other words, the fraction of misclassified points for which the model confidence is larger than the threshold $\tau$ is equal to $\alpha$ on the source. As the spurious correlation decreases and the error rate increases from source to target, the fraction of misclassified points for which the model confidence is larger than the threshold $\tau$ surpasses $\alpha$, leading to a gap in targeted and actual coverage.

Now, we remark on how QTC recalibrates and ensures the target coverage is met. Note that there exists an unknown coverage level $1 - \beta$ that can be used to calibrate TPS on the source distribution such that it yields $1 - \alpha$ coverage on the target. Theorem 6.1 guarantees that QTC correctly estimates $\beta$ and therefore recalibration of the conformal predictor using QTC yields the desired coverage level of $1 - \alpha$ on the target.

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

# Appendix

## A  Proof of Theorem 6.1

In this section, we state and prove a formal version of Theorem 6.1. Our results rely on adapting the proof idea of Garg et al. (2022, Theorem 3) for predicting the classification accuracy of a model to our conformal prediction setup.

Recall that we consider a distribution shift model for a binary classification problem with an invariant predictive feature and a spuriously correlated feature, where a distribution shift is induced by the spurious feature of the target distribution being more or less correlated with the label than the source distribution (Nagarajan et al., 2021; Garg et al., 2022).

We consider a logistic regression classifier that outputs class probability estimates (softmax scores) for the two classes of $y = -1$ and $y = +1$ as

$$(\mathbf{x}) = \left[ \frac{1}{1 + e^{\mathbf{w}^T \mathbf{x}}}, \frac{e^{\mathbf{w}^T \mathbf{x}}}{1 + e^{\mathbf{w}^T \mathbf{x}}} \right],$$

where $\mathbf{w} = [w_{\text{inv}}, w_{\text{sp}}] \in \mathbb{R}^2$. The classifier with $w_{\text{inv}} > 0$ and $w_{\text{sp}} = 0$ minimizes the misclassification error across all choices of distributions $\mathcal{P}$ and $\mathcal{Q}$ (i.e., across all choices of $p$). However, a classifier learned by minimizing the empirical logistic loss via gradient descent depends on both the invariant feature $x_{\text{inv}}$ and the spuriously-correlated feature $x_{\text{sp}}$, i.e., $w_{\text{sp}} \neq 0$ due to the geometric skews on the finite data and statistical skews of the optimization with finite gradient descent steps (Nagarajan et al., 2021).

In order to understand how geometric skews result in learning a classifier that depends on the spurious feature, suppose the probability that the spurious feature agrees with the label is high, i.e., $p$ is close to 1.0. Note that in a finite-size training set drawn from this distribution, the fraction of samples for which the spurious feature disagrees with the label (i.e., $x_{\text{sp}} \neq y$) is small. Therefore, the margin on the invariant feature for these samples alone can be significantly larger than the actual margin $\gamma$ of the underlying distribution. This implies that the max-margin classifier depends positively on the spurious feature, i.e., $w_{\text{sp}} > 0$. Furthermore, we assume that $w_{\text{inv}} > 0$, which is required to obtain non-trivial performance (beating a random guess).

**Conformal prediction in the distribution shift model.**  We consider the conformal prediction method TPS (Sadinle et al., 2019) applied to the linear classifier described above. While other conformal prediction methods such as APS and RAPS also work for this model, the smoothing induced by the randomization of the model scores used in those conformal predictors would introduce additional complexity to the analysis. TPS also tends to be more efficient in that it yields smaller confidence sets compared to APS and RAPS at the same coverage level, see (Angelopoulos et al., 2020, Table 9).

In the remaining part of this section, we establish Theorem 6.1, which states that TPS recalibrated on the source calibration set with QTC achieves the desired coverage of $1 - \alpha$ on any target distribution that has a (potentially) different correlation probability $p$ for the spurious feature. We show this in two steps:

First, consider the oracle conformal predictor that is calibrated to achieve $\alpha$ miscoverage on the target distribution, i.e., the conformal predictor with threshold $\tau_\alpha^{\mathcal{Q}}$ chosen so that

$$\alpha = \mathrm{P}_{(\mathbf{x}, y) \sim \mathcal{Q}} \left[ y \notin \mathcal{C}(\mathbf{x}, \tau_\alpha^{\mathcal{Q}}) \right]. \tag{12}$$

Define the miscoverage on the source distribution as

$$\beta = \mathrm{P}_{(\mathbf{x}, y) \sim \mathcal{P}} \left[ y \notin \mathcal{C}(\mathbf{x}, \tau_\alpha^{\mathcal{Q}}) \right].$$

From those two equations, it follows that a conformal predictor calibrated to achieve miscoverage $\beta$ on the source distribution $\mathcal{P}$ achieves the desired miscoverage of $\alpha$ on the target distribution, provided that the calibration sets are sufficiently large, which is assumed as we consider the case of $n \to \infty$.

Second, we provide a bound on the deviation of the QTC estimate from the true value of $\beta$. We show that in the infinite sample size case, the QTC estimate converges to the true value of $\beta$. Those two steps prove Theorem 6.1.

**Step 1:** QTC relies on the fact that there exists an unknown $\beta \in (0, 1)$ that can be used to calibrate TPS on the source distribution such that it yields $1 - \alpha$ coverage on the target.

Here, we show that callibrating to achieve $1 - \beta$ coverage on the source calibration set $\mathcal{D}^{\mathcal{P}}$ via computing the threshold (2) achieves $1 - \alpha$ coverage on the target distribution $\mathcal{Q}$ as $n \to \infty$.

We utilize the following coverage guarantee of conformal predictors established by Vovk et al. (2005); Lei et al. (2018); Angelopoulos et al. (2020):

**Lemma A.1.** *(Lei et al., 2018, Thm. 2.2), (Angelopoulos et al., 2020, Thm. 1, Prop. 1) Consider* $(\mathbf{x}_i, y_i), i = 1, \ldots, n$ *drawn iid from some distribution* $\mathcal{P}$. *Let* $\mathcal{C}(\mathbf{x}, \tau)$ *be the conformal set generating function that satisfies the nesting property in* $\tau$, *i.e.,* $\mathcal{C}(\mathbf{x}, \tau') \subseteq \mathcal{C}(\mathbf{x}, \tau)$ *if* $\tau' \leq \tau$. *Then, the conformal predictor calibrated by finding* $\tau^*$ *that achieves* $1 - \alpha$ *coverage on the finite set* $\{(\mathbf{x}_i, y)\}_{i=1}^{n}$ *as in (2) achieves* $1 - \alpha$ *coverage on distribution* $\mathcal{P}$, *i.e.,*

$$\mathrm{P}_{(\mathbf{x},y) \sim \mathcal{P}} \left[ y \in \mathcal{C}(\mathbf{x}, \tau^*) \right] \geq 1 - \alpha. \tag{13}$$

*Furthermore, assume that the variables* $s_i = s(\mathbf{x}_i, y_i) = \inf\{\tau : y_i \in \mathcal{C}(\mathbf{x}_i, \tau)\}$ *for* $i = 1, \ldots, n$ *are distinct almost surely. Then, the coverage achieved by the calibrated conformal predictor with the set generating function* $\mathcal{C}(\mathbf{x}, \tau) = \{\ell \in \mathcal{Y} : s(\mathbf{x}, \ell) \leq \tau\}$ *is also accurate, in that it satisfies*

$$\mathrm{P}_{(\mathbf{x},y) \sim \mathcal{P}} \left[ y \in \mathcal{C}(\mathbf{x}, \tau^*) \right] \leq 1 - \alpha + \frac{1}{n+1}. \tag{14}$$

Both the lower bound (13) and the upper bound (14) of Lemma A.1 apply to TPS in the context of the binary classification problem that we consider. To see this, we verify that TPS calibrated with the set generating function (19) satisfies both assumptions of Lemma A.1. First, note that TPS satisfies the nesting property, since we have $\mathcal{C}^{\mathrm{TPS}}(\mathbf{x}, \tau') \subseteq \mathcal{C}^{\mathrm{TPS}}(\mathbf{x}, \tau)$ for $\tau' \leq \tau$. Next, note that for TPS we have $s(\mathbf{x}, y) = \pi_y(\mathbf{x})$. Further note that the linear logistic regression model we consider assigns a distinct score to each data point and since the invariant feature $x_{\mathrm{inv}}$ is uniformly distributed in a continuous interval conditional on $y$, the variables $s_i$ are distinct almost surely.

Now, consider the oracle TPS threshold $\tau_\alpha^{\mathcal{Q}}$ that achieves $1 - \alpha$ coverage, or equivalently $\alpha$ miscoverage, on the target distribution, i.e.,

$$\mathrm{P}_{(\mathbf{x},y) \sim \mathcal{Q}} \left[ y \notin \mathcal{C}^{\mathrm{TPS}}(\mathbf{x}, \tau_\alpha^{\mathcal{Q}}) \right] = \alpha. \tag{15}$$

Next, note that $y \notin \mathcal{C}^{\mathrm{TPS}}(\mathbf{x}, \tau_\alpha^{\mathcal{Q}})$ if and only if $\arg\max_{j \in \{0,1\}} \pi_j(\mathbf{x}) \neq y$ and $\max_{j \in \{0,1\}} \pi_j(\mathbf{x}) \geq \tau_\alpha^{\mathcal{Q}}$. To see that, note that the confidence set returned by TPS is a singleton containing only the top prediction of the model when the confidence of this prediction is higher than the threshold $\tau_\alpha^{\mathcal{Q}}$. Moreover, the confidence set returned by TPS for the binary classification problem above does not contain the true label only when the confidence set is the singleton set of the top prediction of the model and is different than the true label. Thus, equation (15) implies

$$\mathrm{P}_{(\mathbf{x},y) \sim \mathcal{Q}} \left[ \arg\max_{j \in \{0,1\}} \pi_j(\mathbf{x}) \neq y \text{ and } \max_{j \in \{0,1\}} \pi_j(\mathbf{x}) \geq \tau_\alpha^{\mathcal{Q}} \right] = \alpha. \tag{16}$$

We define $\beta$ as the miscoverage that the oracle TPS yields on the source distribution, i.e.,

$$\beta := \mathrm{P}_{(\mathbf{x},y) \sim \mathcal{P}} \left[ \arg\max_{j \in \{0,1\}} \pi_j(\mathbf{x}) \neq y \text{ and } \max_{j \in \{0,1\}} \pi_j(\mathbf{x}) \geq \tau_\alpha^{\mathcal{Q}} \right]. \tag{17}$$

We have $\beta \neq \alpha$ if there is a distribution shift from target to source.

Consider the threshold $\hat{\tau}_\beta^{\mathcal{P}}$ found by calibrating TPS on the set $\mathcal{D}^{\mathcal{P}}$ to achieve empirical coverage of $1 - \beta$ as in (2). TPS with the threshold $\hat{\tau}_\beta^{\mathcal{P}}$ achieves coverage on the source distribution $\mathcal{P}$ as a result of Lemma A.1. Moreover, combining (13) with (14) at $n \to \infty$ yields exact coverage of $1 - \beta$ on the source distribution $\mathcal{P}$. Thus, we have

$$\mathrm{P}_{(\mathbf{x},y) \sim \mathcal{P}} \left[ \arg\max_{j \in \{0,1\}} \pi_j(\mathbf{x}) \neq y \text{ and } \max_{j \in \{0,1\}} \pi_j(\mathbf{x}) \geq \hat{\tau}_\beta^{\mathcal{P}} \right] = \beta. \tag{18}$$

Comparing equation (18) to the definition of $\beta$ in equation (17) yields $\hat{\tau}_\beta^{\mathcal{P}} = \tau_\alpha^{\mathcal{Q}}$. Therefore, it follows that TPS calibrated to achieve $1 - \beta$ coverage on the source calibration set $\mathcal{D}^{\mathcal{P}}$ as in (2) achieves exactly $1 - \alpha$ coverage on the target distribution $\mathcal{Q}$ as $n \to \infty$.

**Step 2:** In the second step, we show that QTC correctly estimates the value of $\beta$ defined above. This is formalized by the lemma below.

Recall that the calibration of TPS entails identifying a cutoff threshold $\tau$ computed by the formula (2). The set generating function of TPS for the linear classification problem described above simplifies to

$$\mathcal{C}^{\mathrm{TPS}}(\mathbf{x}, \tau) = \{j \in \{0, 1\} \colon \pi_j(\mathbf{x}) \geq 1 - \tau\}, \tag{19}$$

where $\pi_0(\mathbf{x})$ and $\pi_1(\mathbf{x})$ are the first and second entry of $(\mathbf{x})$ as defined above.

We are only interested in the regime where the desired coverage level $1 - \alpha$ is larger than the classifier's accuracy, or equivalently $\alpha < \epsilon$ with $\epsilon$ being the error rate of the classifier. This is because a trivial method that constructs confidence sets with equal length of 1 for all samples (i.e., singleton sets of only the predicted label) already achieves coverage of $1 - \epsilon$.

**Lemma A.2.** *Given the logistic regression classifier for the binary classification problem described above with any $w_{\mathrm{inv}} > 0, w_{\mathrm{sp}} \neq 0$, assume that the threshold $q$ for QTC is computed using a dataset $\mathcal{D}^{\mathcal{Q}}$ consisting of $n$ samples, sampled from some target distribution $\mathcal{Q}$, such that*

$$\frac{1}{|\mathcal{D}^{\mathcal{Q}}|} \sum_{\mathbf{x} \in \mathcal{D}^{\mathcal{Q}}} \mathbb{1}_{\{\max_{j \in \{0,1\}} \pi_j(\mathbf{x}) < q\}} = \alpha. \tag{20}$$

*Consider the oracle TPS conformal predictor with conformal threshold $\tau_\alpha^{\mathcal{Q}}$, i.e., the predictor that achieves $1 - \alpha$ coverage on the target distribution $\mathcal{Q}$. Denote with $1 - \beta$ the coverage achieved on the source distribution $\mathcal{P}$ by this oracle TPS. Fix a $\delta > 0$. The QTC estimate of the* miscoverage $\beta$, *denoted by*

$$\beta_{\mathrm{QTC}} = \frac{1}{|\mathcal{D}^{\mathcal{P}}|} \sum_{\mathbf{x} \in \mathcal{D}^{\mathcal{P}}} \mathbb{1}_{\{\max_{j \in \{0,1\}} \pi_j < q\}}, \tag{21}$$

*satisfies the following inequality with probability at least $1 - \delta$ over a randomly drawn set of examples $\mathcal{D}^{\mathcal{Q}}$*

$$|\beta_{\mathrm{QTC}} - \beta| \leq \sqrt{\frac{2 \log(16/\delta)}{n \cdot c_{sp}}}, \tag{22}$$

*where $c_{\mathrm{sp}} = (1 - p^{\mathcal{Q}}) \cdot (1 - p^{\mathcal{P}})^2$ if $w_{\mathrm{sp}} > 0$ and $c_{\mathrm{sp}} = p^{\mathcal{Q}} \cdot (p^{\mathcal{P}})^2$ otherwise.*

*Proof.* We adapt the proof idea of Garg et al. (2022, Theorem 3), which pertains to the problem of estimating the classification error of the classifier on the target, to estimating the source coverage of the oracle conformal predictor that achieves $1 - \alpha$ coverage on the target distribution.

For notational convenience, we define the event that a sample $(\mathbf{x}, y)$ is not in the prediction set of the oracle TPS with conformal threshold $\tau_\alpha^{\mathcal{Q}}$ (i.e., $y \notin \mathcal{C}^{\mathrm{TPS}}(\mathbf{x}, \tau_\alpha^{\mathcal{Q}})$) as

$$\begin{aligned}
\mathcal{E}_{mc} &= \{y \notin \mathcal{C}^{\mathrm{TPS}}(\mathbf{x}, \tau_\alpha^{\mathcal{Q}})\} \\
&= \{\arg \max_{j \in \{0,1\}} \pi_j(\mathbf{x}) \neq y \text{ and } \max_{j \in \{0,1\}} \pi_j(\mathbf{x}) \geq \tau_\alpha^{\mathcal{Q}}\}.
\end{aligned}$$

**The infinite sample size case ($\mathbf{n \to \infty}$).** In this part we show that as $n \to \infty$, the QTC estimate $\beta_{\mathrm{QTC}}$ found as in (21) converges to the source miscoverage $\beta$, to illustrate the proof idea. For $n \to \infty$, the QTC

estimate $\beta_{\text{QTC}}$ in (21) becomes

$$
\begin{aligned}
\beta_{\text{QTC}} &= \mathbb{E}_{(\mathbf{x},y)\sim\mathcal{P}}\left[\mathbb{1}_{\left\{\max_{j\in\{0,1\}} f_j(\mathbf{x})\leq q\right\}}\right] \\
&= \mathrm{P}_{(\mathbf{x},y)\sim\mathcal{P}}\left[\max_{j\in\{0,1\}} \pi_j(\mathbf{x}) < q\right] \\
&= \mathrm{P}_{(\mathbf{x},y)\sim\mathcal{P}}\left[\mathcal{E}_{mc}\right] \\
&= \beta,
\end{aligned}
\tag{23}
$$

where the last equality is the definition of $\beta$ as given in equation (17). The critical step is equation (23), which we establish in the remainder of this part of the proof.

First, we condition on the label $y$. Using the law of total probability, we get

$$
\begin{aligned}
\mathrm{P}_{(\mathbf{x},y)\sim\mathcal{P}}\left[\max_{j\in\{0,1\}} \pi_j(\mathbf{x}) < q\right] &= \mathrm{P}_{\mathbf{x}\sim\mathcal{P}|y=-1}\left[\max_{j\in\{0,1\}} \pi_j(\mathbf{x}) < q\right]\cdot \mathrm{P}_{(\mathbf{x},y)\sim\mathcal{P}}\left[y=-1\right] \\
&\quad + \mathrm{P}_{\mathbf{x}\sim\mathcal{P}|y=+1}\left[\max_{j\in\{0,1\}} \pi_j(\mathbf{x}) < q\right]\cdot \mathrm{P}_{(\mathbf{x},y)\sim\mathcal{P}}\left[y=+1\right] \\
&\stackrel{(i)}{=} \frac{1}{2}\cdot \mathrm{P}_{\mathbf{x}\sim\mathcal{P}|y=-1}\left[\max_{j\in\{0,1\}} \pi_j(\mathbf{x}) < q\right] + \frac{1}{2}\cdot \mathrm{P}_{\mathbf{x}\sim\mathcal{P}|y=+1}\left[\max_{j\in\{0,1\}} \pi_j(\mathbf{x}) < q\right] \\
&\stackrel{(ii)}{=} \mathrm{P}_{\mathbf{x}\sim\mathcal{P}|y}\left[\max_{j\in\{0,1\}} \pi_j(\mathbf{x}) < q\right].
\end{aligned}
\tag{24}
$$

For equation $(i)$, we used that $y$ is uniformly distributed across $\{-1,1\}$, and for equation $(ii)$ that $\mathbf{x}$ is symmetrically distributed with respect to the label $y$. That is, we have $x_{\text{inv}} \sim U[-c,-\gamma]$ and $\mathrm{P}[x_{\text{sp}}=-1]=p$ if $y=-1$ and $x_{\text{inv}}\sim U[\gamma,c]$ and $\mathrm{P}[x_{\text{sp}}=+1]=p$ if $y=+1$, so the two probabilities in $(i)$ are equal.

We can further expand the expression for the probability $\mathrm{P}_{\mathbf{x}\sim\mathcal{P}|y}\left[\max_{j\in\{0,1\}} \pi_j(\mathbf{x}) < q\right]$ by additionally conditioning on the spurious feature $x_{\text{sp}}$, which yields

$$
\begin{aligned}
\mathrm{P}_{(\mathbf{x},y)\sim\mathcal{P}}\left[\max_{j\in\{0,1\}} \pi_j(\mathbf{x}) < q\right] &= \mathrm{P}_{x_{\text{inv}}\sim\mathcal{P}|y,x_{\text{sp}}=y}\left[\max_{j\in\{0,1\}} \pi_j(\mathbf{x}) < q\right]\cdot \mathrm{P}_{\mathbf{x}\sim\mathcal{P}|y}\left[x_{\text{sp}}=y\right] \\
&\quad + \mathrm{P}_{x_{\text{inv}}\sim\mathcal{P}|x_{\text{sp}}\neq y}\left[\max_{j\in\{0,1\}} \pi_j(\mathbf{x}) < q\right]\cdot \mathrm{P}_{\mathbf{x}\sim\mathcal{P}|y}\left[x_{\text{sp}}\neq y\right].
\end{aligned}
\tag{25}
$$

In order to simplify the expression in the RHS of equation (25), we consider the cases of $w_{\text{sp}} > 0$ and $w_{\text{sp}} < 0$ separately. If $w_{\text{sp}} > 0$, we have $\max_{j\in\{0,1\}} \pi_j(\mathbf{x}) > q$ if $x_{\text{sp}} = y$. Therefore, we have $\mathrm{P}_{x_{\text{inv}}\sim\mathcal{P}|y,x_{\text{sp}}=y}\left[\max_{j\in\{0,1\}} \pi_j(\mathbf{x}) < q\right] = 0$ if $w_{\text{sp}} > 0$ and equation (25) simplifies to

$$
\begin{aligned}
\mathrm{P}_{(\mathbf{x},y)\sim\mathcal{P}}\left[\max_{j\in\{0,1\}} \pi_j(\mathbf{x}) < q\right] &= \mathrm{P}_{x_{\text{inv}}\sim\mathcal{P}|x_{\text{sp}}\neq y}\left[\max_{j\in\{0,1\}} \pi_j(\mathbf{x}) < q\right]\cdot \mathrm{P}_{\mathbf{x}\sim\mathcal{P}|y}\left[x_{\text{sp}}\neq y\right] \\
&= \mathrm{P}_{x_{\text{inv}}\sim\mathcal{P}|x_{\text{sp}}\neq y}\left[\max_{j\in\{0,1\}} \pi_j(\mathbf{x}) < q\right]\cdot (1-p^{\mathcal{P}}).
\end{aligned}
\tag{26}
$$

Similarly, if $w_{\text{sp}} < 0$, we have $\max_{j\in\{0,1\}} \pi_j(\mathbf{x}) > q$ if $x_{\text{sp}} \neq y$, and equation (25) becomes

$$
\begin{aligned}
\mathrm{P}_{(\mathbf{x},y)\sim\mathcal{P}}\left[\max_{j\in\{0,1\}} \pi_j(\mathbf{x}) < q\right] &= \mathrm{P}_{x_{\text{inv}}\sim\mathcal{P}|x_{\text{sp}}=y}\left[\max_{j\in\{0,1\}} \pi_j(\mathbf{x}) < q\right]\cdot \mathrm{P}_{\mathbf{x}\sim\mathcal{P}|y}\left[x_{\text{sp}}=y\right] \\
&= \mathrm{P}_{x_{\text{inv}}\sim\mathcal{P}|x_{\text{sp}}=y}\left[\max_{j\in\{0,1\}} \pi_j(\mathbf{x}) < q\right]\cdot p^{\mathcal{P}}.
\end{aligned}
\tag{27}
$$

We next follow the same steps that we carried out above for $\mathrm{P}_{(\mathbf{x},y)\sim\mathcal{P}}\left[\max_{j\in\{0,1\}} \pi_j(\mathbf{x}) < q\right]$ to rewrite the probability $\mathrm{P}_{(\mathbf{x},y)\sim\mathcal{P}}\left[\mathcal{E}_{mc}\right]$. If $w_{\text{sp}} > 0$, the classifier makes no errors if $x_{\text{sp}} = y$ and only misclassifies a fraction

of examples if $x_{\text{sp}} \neq y$. Therefore, we have

$$
\begin{aligned}
\mathrm{P}_{\mathbf{x} \sim \mathcal{P} | y} \left[ \mathcal{E}_{mc} \right] &= \mathrm{P}_{x_{\text{inv}} \sim \mathcal{P} | x_{\text{sp}} \neq y} \left[ \mathcal{E}_{mc} \right] \cdot \mathrm{P}_{\mathbf{x} \sim \mathcal{P} | y} \left[ x_{\text{sp}} \neq y \right] \\
&= \mathrm{P}_{x_{\text{inv}} \sim \mathcal{P} | x_{\text{sp}} \neq y} \left[ \mathcal{E}_{mc} \right] \cdot (1 - p^{\mathcal{P}}).
\end{aligned}
\tag{28}
$$

Similarly, for $w_{\text{sp}} < 0$, we have

$$
\begin{aligned}
\mathrm{P}_{\mathbf{x} \sim \mathcal{P} | y} \left[ \mathcal{E}_{mc} \right] &= \mathrm{P}_{x_{\text{inv}} \sim \mathcal{P} | x_{\text{sp}} \neq y} \left[ \mathcal{E}_{mc} \right] \cdot \mathrm{P}_{\mathbf{x} \sim \mathcal{P} | y} \left[ x_{\text{sp}} = y \right] \\
&= \mathrm{P}_{x_{\text{inv}} \sim \mathcal{P} | x_{\text{sp}} \neq y} \left[ \mathcal{E}_{mc} \right] \cdot p^{\mathcal{P}}.
\end{aligned}
\tag{29}
$$

Therefore, in order to establish equation (23), it suffices to show

$$
\mathrm{P}_{x_{\text{inv}} \sim \mathcal{P} | y, x_{\text{sp}} \neq y} \left[ \max_{j \in \{0,1\}} \pi_j \left( \mathbf{x} \right) < q \right] = \mathrm{P}_{x_{\text{inv}} \sim \mathcal{P} | y, x_{\text{sp}} \neq y} \left[ \mathcal{E}_{mc} \right], \quad \text{for } w_{\text{sp}} > 0 \text{ and}
\tag{30}
$$

$$
\mathrm{P}_{x_{\text{inv}} \sim \mathcal{P} | y, x_{\text{sp}} = y} \left[ \max_{j \in \{0,1\}} \pi_j \left( \mathbf{x} \right) < q \right] = \mathrm{P}_{x_{\text{inv}} \sim \mathcal{P} | y, x_{\text{sp}} = y} \left[ \mathcal{E}_{mc} \right], \quad \text{for } w_{\text{sp}} < 0.
\tag{31}
$$

The feature $x_{\text{inv}}$ is identically distributed conditioned on $y$, i.e., uniformly distributed in the same interval, regardless of the underlying source or target distributions $\mathcal{P}$ and $\mathcal{Q}$. Therefore, equations (30) and (31) are equivalent to

$$
\mathrm{P}_{x_{\text{inv}} \sim \mathcal{Q} | y, x_{\text{sp}} \neq y} \left[ \max_{j \in \{0,1\}} \pi_j \left( \mathbf{x} \right) < q \right] = \mathrm{P}_{x_{\text{inv}} \sim \mathcal{Q} | y, x_{\text{sp}} \neq y} \left[ \mathcal{E}_{mc} \right], \quad \text{for } w_{\text{sp}} > 0 \text{ and}
\tag{32}
$$

$$
\mathrm{P}_{x_{\text{inv}} \sim \mathcal{Q} | y, x_{\text{sp}} = y} \left[ \max_{j \in \{0,1\}} \pi_j \left( \mathbf{x} \right) < q \right] = \mathrm{P}_{x_{\text{inv}} \sim \mathcal{Q} | y, x_{\text{sp}} = y} \left[ \mathcal{E}_{mc} \right], \quad \text{for } w_{\text{sp}} < 0.
\tag{33}
$$

Equations (32) and (33) in turn follow from

$$
\mathrm{P}_{(\mathbf{x}, y) \sim \mathcal{Q}} \left[ \max_{j \in \{0,1\}} \pi_j \left( \mathbf{x} \right) < q \right] = \mathrm{P}_{(\mathbf{x}, y) \sim \mathcal{Q}} \left[ \mathcal{E}_{mc} \right],
\tag{34}
$$

by carrying out the same steps that we carried out to expand the probabilities $\mathrm{P}_{\mathbf{x} \sim \mathcal{P} | y} \left[ \max_{j \in \{0,1\}} \pi_j \left( \mathbf{x} \right) < q \right]$ and $\mathrm{P}_{(\mathbf{x}, y) \sim \mathcal{P}} \left[ \max_{j \in \{0,1\}} \pi_j \left( \mathbf{x} \right) < q \right]$ starting from equation (24) to establish equations (30) and (31). Equation (34) in turn is a consequence of combining (16) with (20) at $n \to \infty$. This establishes equation (23), as desired.

**The finite sample case:** We next show that the desired results approximately hold with high probability over a randomly drawn finite-sized set of examples $\mathcal{D}^{\mathcal{Q}}$. We bound the difference between the LHS and RHS of (32) and (33) with high probability.

First, consider the case of $w_{\text{sp}} > 0$. Recall that for the case of $w_{\text{sp}} > 0$ we are interested in the regime where $x_{\text{sp}} \neq y$. We denote the set of points in the target set $\mathcal{D}^{\mathcal{Q}}$ for which the spurious feature disagrees with the label as

$$
\mathcal{X}_D = \{ i = 1, \ldots, n : x_{\text{sp}} \neq y, (\mathbf{x}_i, y_i) \in \mathcal{D}^{\mathcal{Q}} \},
$$

and denote the set of points for which the spurious feature agrees with the label as

$$
\mathcal{X}_A = \{ i = 1, \ldots, n : x_{\text{sp}} = y, (\mathbf{x}_i, y_i) \in \mathcal{D}^{\mathcal{Q}} \}.
$$

Note that the QTC threshold $q$ found on the entire set $\mathcal{D}^{\mathcal{Q}}$ as in (20) satisfies

$$
\frac{1}{|\mathcal{X}_D|} \sum_{i \in \mathcal{X}_D} \mathbb{1}_{\left\{ \max_{j \in \{0,1\}} \pi_j(\mathbf{x}_i) < q \right\}} = \frac{1}{|\mathcal{X}_D|} \sum_{i \in \mathcal{X}_D} \mathbb{1}_{\{ \mathcal{E}_{mc}(\mathbf{x}_i, y_i) \}},
\tag{35}
$$

which follows from noting that the classifier only makes an error on the subset $\mathcal{X}_D$ if $w_{\mathrm{sp}} > 0$ and therefore the only points for which the event $\mathcal{E}_{mc}$ is observed lie in the set $\mathcal{X}_D$. Similarly, as established before in the infinite sample case, we have $\mathbb{1}_{\left\{\max_{j \in \{0,1\}} \pi_j(\mathbf{x}_i) < q\right\}} = 0$ for all $i \in \mathcal{X}_D$.

By the Dvoretzky-Kiefer-Wolfowitz-Massart (DKWM) inequality, for any $q > 0$ we have with probability at least $1 - \delta/8$

$$\left| \frac{1}{|\mathcal{X}_D|} \sum_{i \in \mathcal{X}_D} \mathbb{1}_{\left\{\max_{j \in \{0,1\}} \pi_j(\mathbf{x}_i) < q\right\}} - \mathbb{E}_{x_{\mathrm{inv}} \sim \mathcal{Q}|y, x_{\mathrm{sp}} \neq y} \left[ \mathbb{1}_{\left\{\max_{j \in \{0,1\}} \pi_j(\mathbf{x}) < q\right\}} \right] \right| \leq \sqrt{\frac{\log(16/\delta)}{2|\mathcal{X}_D|}}. \tag{36}$$

Plugging equation (35) into (36), we have with probability at least $1 - \delta/8$

$$\left| \mathbb{E}_{x_{\mathrm{inv}} \sim \mathcal{Q}|y, x_{\mathrm{sp}} \neq y} \left[ \mathbb{1}_{\left\{\max_{j \in \{0,1\}} \pi_j(\mathbf{x}) < q\right\}} \right] - \frac{1}{|\mathcal{X}_D|} \sum_{i \in \mathcal{X}_D} \mathbb{1}_{\{\mathcal{E}_{mc}\}} \right| \leq \sqrt{\frac{\log(16/\delta)}{2|\mathcal{X}_D|}}. \tag{37}$$

We next bound the second term in the LHS of equation (37) from its expectation. Using Hoeffding's inequality, we have with probability at least $1 - \delta/8$

$$\left| \frac{1}{|\mathcal{X}_D|} \sum_{i \in \mathcal{X}_D} \mathbb{1}_{\{\mathcal{E}_{mc}\}} - \mathbb{E}_{x_{\mathrm{inv}} \sim \mathcal{Q}|y, x_{\mathrm{sp}} \neq y} \left[ \mathbb{1}_{\{\mathcal{E}_{mc}\}} \right] \right| \leq \sqrt{\frac{\log(16/\delta)}{2|\mathcal{X}_D|}}. \tag{38}$$

Combining equations (37) and (38) using the triangle inequality and union bound, we have with probability at least $1 - \delta/4$

$$\left| \mathbb{E}_{x_{\mathrm{inv}} \sim \mathcal{Q}|y, x_{\mathrm{sp}} \neq y} \left[ \mathbb{1}_{\left\{\max_{j \in \{0,1\}} \pi_j(\mathbf{x}) < q\right\}} \right] - \mathbb{E}_{x_{\mathrm{inv}} \sim \mathcal{Q}|y, x_{\mathrm{sp}} \neq y} \left[ \mathbb{1}_{\{\mathcal{E}_{mc}\}} \right] \right| \leq \sqrt{\frac{2 \log(16/\delta)}{|\mathcal{X}_D|}}. \tag{39}$$

Recall that the invariant feature $x_{\mathrm{inv}}$ is uniformly distributed in the same interval conditioned on $y$ regardless of the source or target distributions $\mathcal{P}$ and $\mathcal{Q}$ and that $\mathrm{P}_{x_{\mathrm{inv}}|y, x_{\mathrm{sp}}=y} \left[ \max_{j \in \{0,1\}} \pi_j(\mathbf{x}) > q \right] = \mathrm{P}_{x_{\mathrm{inv}}|y, x_{\mathrm{sp}}=y} \left[ \arg\max_{j \in \{0,1\}} \pi_j(\mathbf{x}) \neq y \right] = 0$ for the case of $w_{\mathrm{sp}} > 0$ as shown before. Therefore, by dividing both sides of (39) with $1/\mathrm{P}_{\mathbf{x} \sim \mathcal{P}|y} \left[ x_{\mathrm{sp}} \neq y \right]$ we have with probability at least $1 - \delta/4$

$$\left| \mathbb{E}_{(\mathbf{x},y) \sim \mathcal{P}} \left[ \mathbb{1}_{\left\{\max_{j \in \{0,1\}} \pi_j(\mathbf{x}) < q\right\}} \right] - \mathbb{E}_{(\mathbf{x},y) \sim \mathcal{P}} \left[ \mathbb{1}_{\{\mathcal{E}_{mc}\}} \right] \right| \leq \frac{1}{\mathrm{P}_{\mathbf{x} \sim \mathcal{P}|y} \left[ x_{\mathrm{sp}} \neq y \right]} \sqrt{\frac{2 \log(16/\delta)}{|\mathcal{X}_D|}}$$

$$= \frac{1}{1 - p^{\mathcal{P}}} \sqrt{\frac{2 \log(16/\delta)}{|\mathcal{X}_D|}}. \tag{40}$$

For the case of $w_{\mathrm{sp}} < 0$, we can show an analogous result by noting that the above results can be shown on the set $\mathcal{X}_A$, where $x_{\mathrm{sp}} = y$. Specifically, noting that $\frac{1}{|\mathcal{X}_A|} \sum_{i \in \mathcal{X}_A} \mathbb{1}_{\left\{\max_{j \in \{0,1\}} \pi_j(\mathbf{x}_i) < q\right\}} = \frac{1}{|\mathcal{X}_A|} \sum_{i \in \mathcal{X}_A} \mathbb{1}_{\{\mathcal{E}_{mc}\}}$ if $w_{\mathrm{sp}} < 0$ and following exactly the same steps from equation (35) onward that lead to equation (40), we have with probability at least $1 - \delta/4$

$$\left| \mathbb{E}_{(\mathbf{x},y) \sim \mathcal{P}} \left[ \mathbb{1}_{\left\{\max_{j \in \{0,1\}} \pi_j(\mathbf{x}) < q\right\}} \right] - \mathbb{E}_{(\mathbf{x},y) \sim \mathcal{P}} \left[ \mathbb{1}_{\{\mathcal{E}_{mc}\}} \right] \right| \leq \frac{1}{p^{\mathcal{P}}} \sqrt{\frac{2 \log(16/\delta)}{|\mathcal{X}_A|}}. \tag{41}$$

Using Hoeffding's inequality we can further bound the RHS of (40) and (41). For the set $\mathcal{X}_D$, we have with probability at least $1 - \delta/2$

$$\left| |\mathcal{X}_D| - n \cdot (1 - p^{\mathcal{Q}}) \right| \leq \sqrt{\frac{\log(4/\delta)}{2n}}, \tag{42}$$

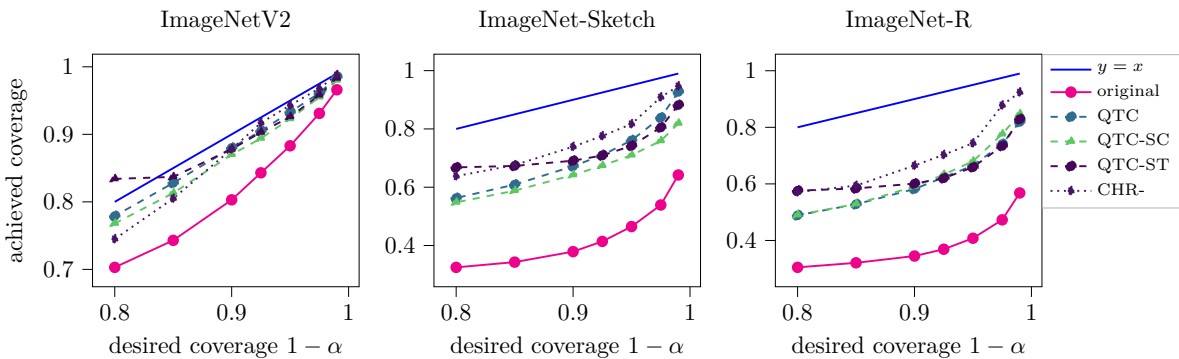

Figure 6: Coverage obtained by RAPS on the target distribution $\mathcal{Q}$ for various settings of $(1 - \alpha)$ w/ and w/o recalibration using QTC.

and for the set $\mathcal{X}_A$, we have with probability at least $1 - \delta/2$

$$\left| |\mathcal{X}_A| - n \cdot p^{\mathcal{Q}} \right| \leq \sqrt{\frac{\log(4/\delta)}{2n}}. \tag{43}$$

We next bound the difference between the finite sample QTC estimation on the source from its expectation. By DKWM inequality, for any $q > 0$ we have with probability at least $1 - \delta/4$

$$\left| \frac{1}{|\mathcal{D}^{\mathcal{P}}|} \sum_{\mathbf{x} \in \mathcal{D}^{\mathcal{P}}} \mathbb{1}_{\left\{ \max_{j \in \{0,1\}} \pi_j(\mathbf{x}) < q \right\}} - \mathbb{E}_{(\mathbf{x}, y) \sim \mathcal{P}} \left[ \mathbb{1}_{\left\{ \max_{j \in \{0,1\}} \pi_j(\mathbf{x}) < q \right\}} \right] \right| \leq \sqrt{\frac{\log(8/\delta)}{2n}}. \tag{44}$$

We first show the result for the case $w_{\text{sp}} > 0$. Combining equations (40) and (44) using the triangle inequality and union bound, we have with probability at least $1 - \delta/2$

$$\left| \frac{1}{|\mathcal{D}^{\mathcal{P}}|} \sum_{\mathbf{x} \in \mathcal{D}^{\mathcal{P}}} \mathbb{1}_{\left\{ \max_{j \in \{0,1\}} \pi_j(\mathbf{x}) < q \right\}} - \mathbb{E}_{(\mathbf{x}, y) \sim \mathcal{P}} \left[ \mathbb{1}_{\{\mathcal{E}_{mc}\}} \right] \right| \leq \frac{1}{1 - p^{\mathcal{P}}} \sqrt{\frac{2 \log(16/\delta)}{|\mathcal{X}_D|}}. \tag{45}$$

Plugging in the definitions of $\beta_{\text{QTC}}$ in (21) and $\beta$ in (17) above, equivalently we get

$$|\beta_{\text{QTC}} - \beta| \leq \frac{1}{1 - p^{\mathcal{P}}} \sqrt{\frac{2 \log(16/\delta)}{|\mathcal{X}_D|}}, \tag{46}$$

which holds with probability at least $1 - \delta/2$. Combining (46) with (42) proves equation (22) for $w_{\text{sp}} > 0$, as desired.

Similarly, for the case $w_{\text{sp}} < 0$, following the same steps by first combining equation (41) with (44), we have with probability at least $1 - \delta/2$

$$|\beta_{\text{QTC}} - \beta| \leq \frac{1}{p^{\mathcal{P}}} \sqrt{\frac{2 \log(16/\delta)}{|\mathcal{X}_A|}}. \tag{47}$$

Combining (47) with (43) yields equation (22), as desired, for the case $w_{\text{sp}} < 0$, which concludes the proof.

$\square$

## A.1 Details on the baseline regression methods

In this section, we provide details on the baseline regression based methods. Recall that we consider several regression-based methods as baselines by fitting a regression function $f_\theta$ parameterized by a feature extractor

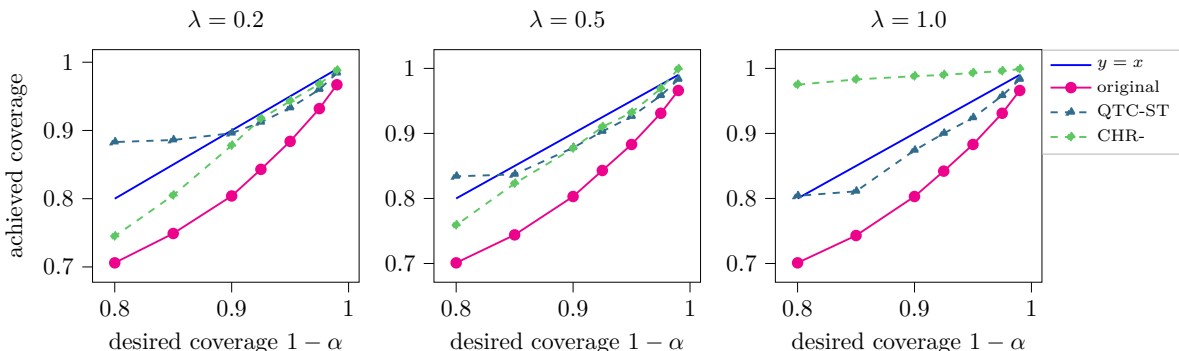

Figure 7: Coverage obtained by RAPS on the target distribution $\mathcal{Q}$ (ImageNetV2) for $k_{\text{reg}} = 2$ and various settings of $\lambda$ when the threshold $\tau$ is replaced with the predicted threshold $\hat{\tau}$ with the respective prediction method. For regression methods, only the best performing method of CHR- is shown.

$\phi_\pi$ by minimizing the mean squared error between the output and the calibrated threshold $\tau$ across the distributions as

$$\hat{\theta} = \arg\min_\theta \sum_j (f_\theta(\phi_\pi(\mathcal{D}_j)) - \tau^{\mathcal{P}_j})^2.$$

We consider the following choices for the feature extractor $\phi_\pi$:

- *Average confidence regression (ACR)*: The one-dimensional ($d = 1$) average confidence of the classifier across the entire dataset which is $\phi_\pi(\mathcal{D}) = \frac{1}{|\mathcal{D}|} \sum_{\mathbf{x} \in \mathcal{D}} \max_\ell \pi_\ell(\mathbf{x})$.

- *Difference of confidence regression (DCR)* (Guillory et al., 2021): The one-dimensional ($d = 1$) average confidence of the classifier across the entire dataset offset by the average confidence on the source dataset, which is $\phi_\pi(\mathcal{D}) = \frac{1}{|\mathcal{D}|} \sum_{\mathbf{x} \in \mathcal{D}} \max_\ell \pi_\ell(\mathbf{x}) - \frac{1}{|\mathcal{D}^{\mathcal{P}}|} \sum_{\mathbf{x} \in \mathcal{D}^{\mathcal{P}}} \max_\ell \pi_\ell(\mathbf{x})$, where $\mathcal{D}^{\mathcal{P}}$ is the source dataset. Prediction is also for the offset target $\tau - \tau^{\mathcal{P}}$.

  We consider DCR in addition to ACR, because DCR performs better for predicting the classifier accuracy (Guillory et al., 2021). Since the threshold $\tau$ found by conformal calibration depends on the distribution of the confidences beyond the average, we propose the below techniques for extracting more detailed information from the dataset.

- *Confidence histogram-density regression (CHR)*: Variable dimensional ($d = p$) features extracted as $\phi_\pi(\mathcal{D}) = \left\{ \frac{1}{|\mathcal{D}|} \sum_{\mathbf{x} \in \mathcal{D}} \mathbb{1}_{\left\{ \max_\ell \pi_\ell(\mathbf{x}) \in \left[ \frac{j-1}{p}, \frac{j}{p} \right] \right\}} \right\}_{j = \{1, \ldots, p\}}$. This corresponds to the normalized histogram density of the classifier confidence across the dataset, where p is a hyperparameter that determines the number of histogram bins in the probability range $[0, 1]$. Neural networks tend to be overconfident in their prediction which heavily skews the histogram densities to the last bin. We also therefore consider a variant of CHR, *dubbed CHR-*, where we have $j = \{1, \ldots, p - 1\}$ and hence $d = p - 1$, equivalent to dropping the last bin of the histogram as a feature.

- *Predicted class-wise average confidence regression (PCR)*: Features with dimensionality equal to the number of classes ($d = L$) extracted as $\phi_\pi(\mathcal{D}) = \left\{ \frac{\sum_{\mathbf{x} \in \mathcal{D}} \pi_j(\mathbf{x}) \cdot \mathbb{1}_{\{l = \arg\max_\ell \pi_\ell(\mathbf{x})\}}}{\sum_{\mathbf{x} \in \mathcal{D}} \mathbb{1}_{\{l = \arg\max_\ell \pi_\ell(\mathbf{x})\}}} \right\}_{j = \{1, \ldots, L\}}$. This corresponds to the average confidence of the classifier across the samples for each predicted class.

## B  RAPS recalibration experiments

APS is a powerful yet simple conformal predictor. However, other conformal predictors (Sadinle et al., 2019; Angelopoulos et al., 2020) are more efficient (in that they have on average smaller confidence sets for a given desired coverage $1 - \alpha$).

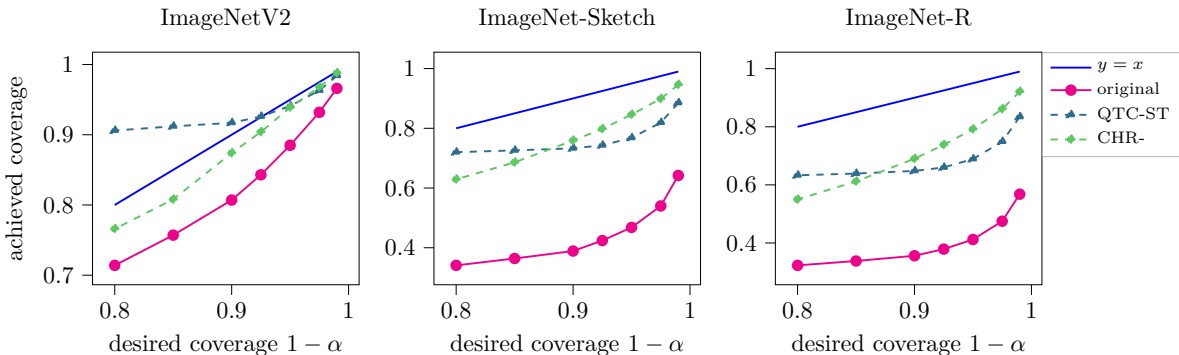

Figure 8: Coverage obtained by RAPS on the target distribution $\mathcal{Q}$ for $\lambda = 0.1$ and various settings of $(1 - \alpha)$ when the threshold $\tau$ is replaced with the predicted threshold $\hat{\tau}$ with the respective prediction method. For regression methods, only the best performing method of CHR- is shown.

In this section, we focus on the conformal predictor proposed by Angelopoulos et al. (2020), dubbed Regularized Adaptive Prediction Sets (RAPS). RAPS is an extension of APS that is obtained by adding a regularizing term to the classifier's probability estimates of the higher-order predictions (i.e., subsequent predictions after the top-k predictions). RAPS is more efficient and tends to produce smaller confidence sets when calibrated on the same calibration set as APS, as it penalizes large sets. While TPS tends to achieve slightly better results in terms of efficiency compared to RAPS, see (Angelopoulos et al., 2020, Table 9), RAPS coverage tends to be more uniform across different instances (in terms of difficult vs. easy instances) and therefore RAPS still carries practical relevance.

Recall that while efficiency can be improved by constructing confidence sets more aggressively, efficient models tend to be less robust, meaning the coverage gap is greater when there is distribution shift at test time. For example, when calibrated to yield $1 - \alpha = 0.9$ coverage on ImageNet-Val and tested on Image-Sketch, the coverage of RAPS drops to 0.38 in contrast to that of APS, which only drops to 0.64 (see Section 3). It is therefore of great interest to understand how QTC performs for recalibration of RAPS under distribution shift.

RAPS is calibrated using exactly the same conformal calibration process as APS and only differs from APS in terms of the prediction set function $\mathcal{C}(\mathbf{x}, u, \tau)$. The prediction set function for RAPS is defined as

$$\mathcal{C}^{\text{RAPS}}(\mathbf{x}, u, \tau) = \left\{ \ell \in \{1, \ldots, L\} \colon \sum_{j=1}^{\ell-1} [\pi_{(j)}(\mathbf{x}) + \underbrace{\mathbb{1}_{\{j - k_{\text{reg}} > 0\}} \cdot \lambda}_{\text{regularization}}] + u \cdot \pi_{(\ell)}(\mathbf{x}) \leq \tau \right\}, \tag{48}$$

where $u \sim U(0, 1)$, similar to APS and $\lambda, k_{\text{reg}}$ are the hyperparameters of RAPS corresponding to the regularization amount and the number of top non-penalized predictions respectively.

Note that the cutoff threshold $\tau^{\mathcal{P}}$ obtained by calibrating RAPS on some calibration set $\mathcal{D}_{\text{cal}}^{\mathcal{P}}$ can be larger than one due to the added regularization. Therefore, in order to apply QTC-ST, we map $\tau^{\mathcal{P}}$ back to the $[0, 1]$ range by dividing by the total scores after added regularization. QTC and QTC-SC do not require such an additional step as the coverage level $\alpha \in [0, 1]$ by definition. We show the results of RAPS' performance under distribution shift with or without calibration by QTC in Figure 6. The results show that while QTC is not able to completely mitigate the coverage gap, it significantly reduces it.

Recall that RAPS utilizes a hyperparameter $\lambda$, which is the added penalty to the scores of the predictions following the top-$k_{\text{reg}}$ predictions, that can significantly change the cutoff threshold $\tau^{\mathcal{P}}$ when we calibrate on the calibration set $\mathcal{D}^{\mathcal{P}}$. The regularization amount $\lambda$ also implicitly controls the change in the cutoff threshold $\left| \tau^{\mathcal{Q}} - \tau^{\mathcal{P}} \right|$ when the conformal predictor is calibrated on different distributions $\mathcal{P}$ and $\mathcal{Q}$. That is, the value of $\left| \tau^{\mathcal{Q}} - \tau^{\mathcal{P}} \right|$ increases with increasing $\lambda$ as long as the distributions $\mathcal{P}$ and $\mathcal{Q}$ are meaningfully different, as is the case for all the distribution shifts that we consider.

Therefore, a good recalibration method should be relatively immune to the choice of $\lambda$ in order to successfully predict the threshold $\tau^{\mathcal{Q}}$ based only on unlabeled examples. In Figure 7, we show the performance of RAPS under the ImageNetV2 distribution shift for various values of $\lambda$. While QTC is able to improve the coverage gap for various choices of $\lambda$, the best performing regression-based baseline method does not generalize well to natural distribution shifts when $\lambda$ is relatively large. In contrast, as demonstrated in Figure 8, when the regularization amount $\lambda$ is relatively small, the best performing regression-based method of CHR- does very well in reducing the coverage gap of RAPS under various distribution shifts.

