# OpenReview forum: "Test-time recalibration of conformal predictors under distribution shift based on unlabeled examples"
_TMLR — Rejected by TMLR_

### Review · Reviewer_u1aw · 2024-01-15

**Summary Of Contributions:**

The paper focuses on out-of-distribution testing. The authors propose a recalibration approach to improve the performance of conformal prediction under distribution shifts. The recalibration adjustment is obtained from unlabelled samples from the target distribution. Intuitively, the idea is to replace the nominal coverage with an adjusted value when computing the sample quantile of the calibration conformity scores. The adjusted value is (a function of) the empirical coverage obtained on the calibration set with the quantile estimated on the test set and vice-versa.

**Audience:**

Yes

**Broader Impact Concerns:**

No concerns.

**Claims And Evidence:**

Yes

**Requested Changes:**

- Why for  *image datasets covariate shift is not well defined*?
- In Equations 7 and 8, could BTC-S and BTC-T be defined in the same way (i.e. just flipping $D_{cal}$ and $D_{test}$)? If not, is there an intuitive reason why using $\alpha$ and $1 - \alpha$?
- What are the assumptions under which *there is a value $\beta$ such that, if we calibrate the conformal predictor on the source distribution using the value $\beta$ instead of $\alpha$, it achieves $1 - \alpha$ coverage on the target distribution*?
- What happens if the classifier is bad and the ${\rm \arg} \max \pi$ is often different from the label?
- How does the synthetic distribution affect the baseline's performance?
- It would be interesting to see QTC-T and QTC-S in the plots.
- Why do covariate-shift methods *rely on labelled examples from the target domain*?
- Why does PS-W always produce too-big intervals (over-coverage) while the other methods always produce too-small intervals (under-coverage)? Is PS-W the only method with theoretical coverage guarantees? If so, is it fair to compare with it?
- It looks like the performance of WSCI depends on how the discriminator is trained. Have you tried different strategies in the one-domain scenario?

**Strengths And Weaknesses:**

**Strengths**
The validity of Conformal Prediction under distribution shift is a timely problem. The idea of using unlabelled samples from the target distribution is interesting. The method is general and produces good prediction sets on real-world data sets.

**Weaknesses**
*Motivations and clarity.* The authors may motivate their work better by providing a practical example of when their method may be better than others. I found it hard to understand the estimation method from the second paragraph of Section 4.1. It would be helpful to add a pseudocode of the procedure. The authors may help the reader by explaining intuitively why their method is expected to produce good prediction sets.
*Theoretical guarantees.* The authors provide asymptotic validity for one method but use another in the experiments. Its performance is not even shown in the experiments. Validity is proven in the limit of an infinite sample size and for a specific distribution. This contradicts CP's finite-sample and distribution-free spirit and may limit the method's impact.

---

> ### Author Response · Authors · 2024-02-13
> **Authors' Response**
>
> Thanks for the valuable feedback and for noting the importance of the problem we tackle.
>
> &nbsp;
>
> - Regarding the **theoretical results being provided for a different method**, we provide theoretical results for QTC-S as it is used in the experiments. Fig. 2 shows the performance of QTC-S across all datasets for a fixed $\alpha$.
>
> - Regarding **why covariate shift is not well defined for image datasets**: A co-variate shift is typically defined in a generative model, and the co-variates of that generative model shift. It's not clear how that is a good model for an application such as image classification. A covariate shift as it is defined (i.e., $p_{\mathcal P} (\mathbf{x}) \neq p_{\mathcal P} (\mathbf{x})$) is not a good model when $\mathbf{x}$ are the image pixels since change in the $\mathbf{x}$ does not necessarily capture the change in the abstract features. One possible alternative is to use the embeddings of a deep neural network trained on images to extract useful features mapped to a lower dimensional space to measure the covariate shift. However, the resulting performance tends to be poor based on our observations for existing covariate shift based methods.
>
> - Regarding the definitions of QTC-T and QTC-S in Eq. (7) and (8), the **two methods cannot be simply flipped without also replacing the $\alpha$ and $1-\alpha$s**. The reason for this is the implicit knowledge of the relationship between the model’s scores ($s(\pi (\mathbf x))$ and accuracy. In all of our experiments, and as shown in the literature on the calibration of neural networks’ scores (see Fig. 1 from both [Guo et al., (2017)](https://proceedings.mlr.press/v70/guo17a.html) and [Yu et al., (2022)](http://arxiv.org/abs/2206.02757)), there typically is a strict monotonous relationship between the model scores and accuracy across the dataset, and if the accuracy dropped, the scores also drop. However, the reduction in the scores does not fully capture the reduction in the accuracy when there is a distribution shift, leading to coverage gap. QTC works well in improving the coverage gap as it is able to leverage the change of the scores in relation to a calibrated threshold across the source and target.
> If there exists a **scenario where we have a bad classifier that yields lower scores, but higher accuracy**, yes, then QTC would fail by producing unnecessarily large confidence sets (over-coverage).
>
> - The assumptions for the **existence of a $\beta$ such that, if we calibrate the conformal predictor on the source distribution using the value $\beta$ instead of $\alpha$, it achieves $1 - \alpha$ coverage on the target distribution** is that the distribution of the model scores should be smooth (in addition to being unique as required by conformal prediction, see Lemma A.1) in the $[1 - \alpha - \delta, 1 - \alpha + \delta]$ regime such that a small change in $\beta \in [1 - \alpha - \delta, 1 - \alpha + \delta]$ will change the value of the $\beta$-quantile. In practice, we observed that for all the natural datasets that we consider in our experiments, the distribution of the model scores is sufficiently smooth to identify $\beta$ with an error that scales with $1/n$ (i.e. similar to $\alpha$ for conformal prediction).
>
> - Regarding the **individual results for QTC-S and QTC-T in the plots**, we provide it in Fig. 2 for the bar plot, but omit it for the curves in other plots for the sake of clarity. To provide more context, the relationship between QTC-S and QTC-T presented in Fig. 2 consistently holds across for all x in the remaining line plots for the same dataset.
>
> - Normally, **covariate shift methods do not rely on labeled examples from the target domain**. However, since covariate shift is not well defined for image datasets and there is no good way to measure the amount of shift for image datasets, the only application of covariate shift based methods for image tasks that we know of rely on training a separate discriminator that needs to train on labeled samples from all target domains in order to calculate the importance weights.
>
> - Regarding **why PS-W always produces too-big intervals**, both WSCI and PS-W theoretically guarantee coverage in the same way as long as importance weights are known/can be accurately calculated. Note that PS-W includes all possible classes in the confidence sets for $1 - \alpha \in [0.9, 1]$ and thus fails to quantify any uncertainty. This is not possible to know a priori since the experiments in the original paper only consider $1 - \alpha = 0.8$ and demonstrate better performance than WSCI.

---

> > ### Author Response · Authors · 2024-02-13
> > **Authors' Response (cont'd)**
> >
> > - Regarding **how the trainer is trained for WSCI**, we trained the discriminator following [Park et al., (2022)](https://openreview.net/pdf?id=DhP9L8vIyLc) by using all labeled samples from all source and target domains, but limited the source dataset to a single domain for the classifier only in the case of the one-domain scenario. We have not tried limiting the dataset the discriminator was trained on to only the source (same one domain) and target domains as it would be more logical to directly calibrate the conformal predictor if we already know what the source and target domains are.

---

### Review · Reviewer_Ky8c · 2024-01-18

**Summary Of Contributions:**

The submission studies the problem of Conformal Prediction (CP) under distribution shift when:
* A labeled calibration set is available, but
* Only unlabeled data from the test distribution are observed.

Akin to the Average Thresholded Confidence (ATC) (Garg et al., 2022), the proposed method tries and predict a miscoverage level $\beta$ such that providing $1 - \beta$ coverage on the calibration set implies $1 - \alpha$ coverage on the test distribution.

Experiments are included to compare with regression and covariate-shift based approaches on several imaging datasets with natural distribution shifts.

Finally, a synthetic data experiment from Garg et al. is revisited to provide theoretical consistency guarantees on the proposed method.

**Audience:**

Yes

**Claims And Evidence:**

No

**Requested Changes:**

**Related works**

There are a few works that study CP beyond exchangeability or under covariate shift that are not mentioned in the text. For example:
1. Barber et al., 2023, Conformal Prediction Beyond Exchangeability
2. Prinster et al., 2022, JAWS: Auditing Predictive Uncertainty Under Covariate Shift
3. Prinster et al., 2023, JAWS-X: Addressing Efficiency Bottlenecks of Conformal Prediction Under Standard and Feedback Covariate Shift
4. Gibbs et al., 2021, Adaptive Conformal Inference Under Distribution Shift
5. Gibbs et al., 2023, Conformal Inference for Online Prediction with Arbitrary Distribution Shifts
6. Fannjiang et al., 2022, Conformal prediction under feedback covariate shift for biomolecular design

---

**Questions on proposed method**

- Existence and uniqueness of $\beta$. What is the claim here? Does $\beta$ always exist? Is it unique? Are there any assumptions on the distribution of the data and the predictor? In particular, I am not sure I follow the claim in Appendix A that Eq. (17) and (18) yield $\hat{\tau}^P_\beta = \tau^Q_\alpha$. The equations read to me as $P_{(x, y) \sim P}[A(x,y) \wedge (C(x) \geq \tau)] = P_{(x, y) \sim P}[A(x,y) \wedge (C(x) \geq \tau')]$. However, this does not imply $\tau = \tau'$ in the most general case. For example, consider the case where $C(x) \in [0.5, 1]$ and $P[C(x) \in [0.6, 0.7]] = 0$. Then, if the LHS holds for $\tau = 0.65$, the RHS also holds for any $\tau' \in (0.6, 0.7)$. Am I missing something here?

- $\beta_{QTC}$ in Eq. (6) versus $\beta_{QTC}$ in Eq. (21): The theoretical results only apply to $\beta_{QTC-T}$ as presented in Eq. (7). Besides making sure the definitions are consistent throughout the paper, I would be curious to know whether there are bottlenecks on proving similar consistency results for $\beta_{QTC-S}$?

- Motivation and discussion of $\beta_{QTC}$ in Eq. (6): The current presentation of the proposed method might improve with a discussion of what the quantities represent. For example, it might help readers know what Eq. (7) and (8) are when there is no distribution shift.

- I am also wondering why the threshold in $\beta_{QTC-T}$ is defined on the test set, whereas it is defined on the calibration set for $\beta_{QTC-S}$. This is not immediately clear to me and I would love to hear the authors ideas and intuition.

---

**Experimental results**

- Figure 2: It might be useful to include the average set sizes and the value of the estimated $\beta$ to show that QTC remains efficient. It is true QTC seems to be closing the gap in coverage, but it almost always fails to provide the desired level of coverage.

- Figure 2 vs Figure 3: Is there any reason why APS is not included in the results presented in Figure 2? From Figure 3, it seems like APS with QTC does provide coverage on ImageNet? Whereas TPS does not.

- Figure 3: It may improve visualization to make the axis equal such that the diagonal is centered in the plots.

- Synthetic experiment: It could be useful to have a synthetic experiment where the true value of $\beta$ is known, to show that the estimate does converge. Including a plot with coverage as a function of $n$ in Figure 5 might be a good way to display the theoretical results in practice, especially because QTC provably works in this setting.

---

**Minor comments**
- Introduction: repetitions in "generate valid set generating [...] a calibration set"
- Related work: typo in "that assume an covariate shift"
- Background: repetitions in "based on calibrating on a calibration set", typo in "the calibration set set was sampled from"
- Failures under distribution shifts: incomplete sentence in "coverage on the only achieves a coverage of 0.64"
- Paragraph after Theorem 6.1: why does TPS only provide asymptotic coverage as $n \to \infty$ here?
- Appendix A:
    - Missing $f$ in first equation.
    - Is the paragraph "From those two equations ... the case of $n \to \infty$" correctly placed here?
    - Typos "misscoverage" and "callibrated".
    - The set generating function (i.e., Eq. (19)) is cited in the text much earlier than it is introduced. Moving would increase readability.
    - Missing $f$ after Eq. (19).
    - Eq. (20), the threshold presented here differs from the quantile definition in Eq. (5).
    - Eq. (21), what is $s$ here? I assume it is the max as throughout the Appendix.

I am looking forward to discussing with the authors and clarify my questions!

**Strengths And Weaknesses:**

Strenghts:
* The problem of CP beyond the exchangeability assumption is relevant
* The case where no labels are observed for the test distribution is compelling and non-trivial
* The proposed method is novel and intriguing

Weaknesses:
* The motivation and presentation of the method is sometimes hand-wavy
* Somewhat limited empirical evidence of the effectiveness of the method

I will expand on these points and I am looking forward to engaging with the authors to clarify my questions.

---

> ### Author Response · Authors · 2024-02-13
> **Authors' Response**
>
> Thanks for the review and the valuable feedback and for noting the importance of the problem and setting. We hope to address the comments below.
>
> &nbsp;
>
> **Questions on the proposed method**
> - Regarding existence and uniqueness of $\beta$, our main assumption is that the distribution of the model scores should be smooth (in addition to being unique as required by conformal prediction, see Lemma A.1) in the $[1 - \alpha - \delta, 1 - \alpha + \delta]$ regime such that a small change in $\beta \in [1 - \alpha - \delta, 1 - \alpha + \delta]$ will change the value of the $\beta$-quantile. In practice, we observed that for all the natural datasets that we consider in our experiments, the distribution of the model scores is sufficiently smooth to identify $\beta$ with an error that scales with $1/n$ (i.e., similar to $\alpha$ for conformal prediction). We agree that QTC will not work well in a scenario where there is a sharp discontinuity in the probability distribution. Moreover, note that $\tau$ itself is determined using a quantile (see Eq. (2)) and we always assume the smallest $\tau$ that satisfies the assumptions under both source and target distributions.
>
> - Regarding whether there are bottlenecks on proving similar theoretical consistency results for $\beta_{\mathrm{QTC-S}}$, yes, in general it is not possible to estimate the coverage successfully with QTC-S even in the toy model. The problem arises due to the fact that $1 - \alpha$ is much larger than the classification error. Therefore, the corresponding quantiles include samples in both sets ${{\mathcal X}_D}$ and ${{\mathcal X}_A}$ defined above Eq. (35). Since the two partitions shift differently under the distribution shift due to the spurious feature, while we can technically prove that the performance is non-decreasing with QTC-S, we can’t prove guaranteed coverage.
>
> - Regarding why the threshold for QTC-T is defined on the test set vs. that of QTC-S which is defined on the source, this is because of the change in direction due to the dependence on $\alpha$ vs. $1 - \alpha$. If the threshold was defined on the source instead, the resulting $\beta$ would be higher than the $\alpha$ when we test on a more difficult distribution for the trained classifier since we expect the model scores to decrease and result in a larger portion of the samples to be under the threshold (i.e., $s (\pi (\mathbf x)) < q$). Instead, we want $1 - \beta$ to increase. Therefore, QTC-S, which uses $1 - \alpha$ (and $1 - \beta$) is defined on the source and QTC-T, which uses $\alpha$ (and $\beta$) is defined on the target.
>
> - Regarding the motivation and discussion of $\beta_{\mathrm{QTC}}$ in Eq. (6), we have added a couple of paragraphs at the end of Step 1 under Sec. 4.1 to clarify the setting.
>
> **Experimental results**
> - Regarding **the exclusion of the set sizes from the experiments**, we omitted the set sizes because they are not informative for QTC (and any other methods we introduce). That is because, since QTC does not alter the underlying conformal predictor, there is a monotonic relation between the coverage and set sizes. If the coverage of two methods (including the vanilla CP) is the same, the set sizes are the same, averaged across any randomness. Moreover, if the coverage is lower then the set size is smaller, and if the coverage is higher, then the set size is larger throughout. Note that the set size becomes pertinent when different conformal predictors are compared, such as in Fig. 4, for which we provide the set sizes. We are happy to add set sizes to all the figures if that would be helpful nevertheless.
>
> - Regarding **the exclusion of APS from Fig. 2**, this is relevant to the set sizes problem. That is, TPS is more efficient than APS, but much less robust to distribution shift (0.34 vs. 0.64 coverage for ImageNet-Sketch, see Sec. 3; Par. 2). Therefore, we believe it’s much more important that QTC is able to significantly close the coverage gap even for TPS. QTC indeed performs much better (in terms of remaining coverage gap) when using APS.
>
> - Regarding the scales of the subplots in Fig. 3, we thought it could be misleading to have different scales across the subplots for the same methods. We didn’t want to include another version of the same plots in the appendix either due to the repetition without much additional information. We would be happy to include that if that would help with the presentation.
>
> - Regarding a synthetic experiment where the true value of $\beta$ is known, we have revised the supplement to add a notebook with the code to recreate the toy model that we consider in the theory section ([also can be accessed here](https://www.dropbox.com/scl/fo/n8s82b8t7tppz92u15zcd/h?rlkey=hsu9mp05lgofld8l2gvu4ury3&dl=0)). Based on our observations, QTC converges with $n > 100$ samples.

---

> > ### Author Response · Authors · 2024-02-13
> > **Authors' Response (cont'd)**
> >
> > **Other comments**
> > - Thanks for bringing the related works to our attention. We are now citing them.
> >
> > - Regarding why TPS only provides asymptotic coverage as $n \rightarrow \infty$ in the paragraph after Theorem 6.1, conformal prediction guarantees $> 1 - \alpha$ coverage, but converges to exactly $1 - \alpha$ as $n \to \infty$ (see Eq. (14) in Lemma A.1)
> >
> > - Regarding why the threshold in Eq. (20) differs from that of in Eq. (5), Eq. (20) assumes that the threshold $q$ is already found using QTC-T, i.e., $q = q({\mathcal D}^{\mathcal Q}, \alpha)$ using Eq. (5), with the simplified notation for the binary case.
> >
> > - Thanks a lot for flagging the typos and errors, we have revised the manuscript to fix them.

---

### Review · Reviewer_TWkd · 2024-01-18

**Summary Of Contributions:**

The paper proposes a simple strategy for calibrating the threshold of conformal predictors under distribution shift. The method relies on calculating the quantiles of an unlabeled set and is easy to compute. The results are compared to a regression-based baseline.

**Audience:**

Yes

**Claims And Evidence:**

Yes

**Requested Changes:**

- Please provide a more intuitive explanation of the approach. Currently, the method is described via a couple of formulas that fail to provide an intuitive understanding of the technique.
- Please consider alternative baselines from related work.
- Minor typos: page 2, "an covariate" -> "a covariate", page 3, use \citep for Romano et al. (2020), also page 3, "we're" -> "we are", also page 3, "coverage on the only achieves" -> "coverage only achieves"

**Strengths And Weaknesses:**

### Strengths
- The paper is well organized, and the problem description is clear.
- The method is quite simple yet effective.

### Weaknesses
- The authors do not provide intuition about the way the proxy threshold $\beta$ is calculated. A reader cannot extract much information only based on the formulas (5)-(8). What is lacking is an intuitive explanation of what each of these elements is. For instance, a pictorial explanation of what the quantile (5) and the thresholds (7) and (8) mean would be highly beneficial. Figure 1 doesn't provide any extra information than what is already presented in the text.
- I am not an expert in the field, and I am not quite sure if there are any standard baselines other than the regression-based method, especially given that the regression-based method is proposed by the authors. I encourage the authors to compare to possibly different well-established baselines or alternative ways of setting the threshold (based on intuitive heuristics).
- What would QTC yield when the source and the target distributions are identical? This would be another way of validating the approach.

---

> ### Author Response · Authors · 2024-02-13
> **Authors' Response**
>
> Thanks for the review and the valuable feedback.
>
> &nbsp;
>
> - Regarding a **more intuitive explanation of the approach**, the intuition for QTC is as follows.
>
> If we had sufficient labeled target data (which we do not have in our setup), then we would compute the (oracle) threshold for conformal calibration as ($\tau^\prime = \tau_\alpha^{\mathcal Q}$ for all equations below; used due to rendering engine fault)
>
> $$
> \tau_\alpha^{\mathcal Q} \colon {\mathrm P}_{({\mathbb x},y)\sim {\mathcal Q}}
> [y \notin {\mathcal C}({\mathbb x},\tau^\prime)] = \alpha
> $$
>
> Instead we compute a threshold $q^{\mathcal Q}$ based on unlabeled data so that
>
> $$
> q^{\mathcal Q} \colon {\mathrm P}_{{\mathbb x} \sim {\mathcal Q}}[
> s(\pi({\mathbb x})) < q^{\mathcal Q}
>  ] = \alpha.
> $$
>
> We then compute the miscoverage on the source distribution ${\mathcal P}$ as
> $$
> \beta_{\mathrm{QTC}}
> = {\mathrm P}_{{\mathbb x} \sim {\mathcal P}}[
> s(\pi({\mathbb x})) < q
> ]
> $$
>
> and if this is equal to
> $$
> \beta
> = {\mathrm P}_{({\mathbb x},y)\sim {\mathcal P}}[
> y \notin {\mathcal C}({\mathbb x},\tau^\prime)
> ]
> $$
>
> then QTC succeeds at re-calibration. Thus, QTC relies on the condition
> $$
> \begin{align}
> {\mathrm P}_{{\mathbb x} \sim {\mathcal P}} [
> s(\pi({\mathbb x})) < q^{\mathcal Q} ]
> \end{align}
> $$
>
> $$
> \begin{align}
> \approx
> {\mathrm P}_{({\mathbb x},y) \sim {\mathcal P}} [
> y \notin {\mathcal C} ({\mathbb x}, \tau^\prime)]
> \end{align}
> $$
>
>
> to be (at least approximately) true.
> We have found this relation to be approximately true for a variety of distribution shifts in practice and our theory section shows that there are distributions shifts where this relation is true and where QTC provably succeeds.
>
> - Regarding the **lack of other baselines to compare against**, we compare to regression-based algorithms and to covariate shift based methods for the following reasons.
> We were unable to find any methods that can calibrate conformal predictors with no labeled target data. While we compare to a couple covariate shift based methods in Sec. 5.2 that don’t necessarily require labeled data, note that for dataset where the covariate shift is not well defined or measured, those methods rely on the existence of a discriminator model that is trained on labeled data to classify the target distribution from that of the training/calibration. We therefore needed to establish additional baselines that do not require any labeled data for training.
>
> - Thanks a lot for flagging the typos, we have revised the manuscript to fix them.

---

### Decision · Action_Editor_dupu · 2024-03-21

**Recommendation:** Reject

**Comment:**

The reviewers found that the authors tackled an important and challenging problem and the idea is novel and interesting.  However, Reviewer Ky8c pointed out that the presentation is hand-wavy and results do not provide enough evidence towards the effectiveness of the proposed method.  I quote the message by Reviewer Ky8c, which should be well-addressed for future acceptance (I encourage resubmission).  In general, the authors should revise the paragraphs on which reviewers asked questions, because it implies that something is unclear in the text.


***********

- The motivation behind the choice of estimators remains unclear. A brief paragraph was added to Sec. 4.1, but this is the main contribution of the submission, and I feel it should be introduced more carefully.
- Theoretical results to motivate the choice of estimators remain unclear. Both Reviewer u1aw and I asked for the assumptions behind the validity of the theoretical results. This question has been answered partially in our discussion with the authors, but no mentions of such assumptions have been added to the text.
- Theoretical results only apply to QTC-T. In my discussion with the authors, they mentioned the hardness of proving similar results for QTC-S. If this is the case, then the use of QTC-S needs to be motivated differently, because there are no theoretical results in support of why it should work.
- Experimental evidence remains limited. QTC does not always achieve the desired coverage, and it is unclear where these failure cases come from: whether the underlying assumptions of the method are not satisfied, or some other reason. It is unclear to me whether QTC outperforms other methods simply because it constructs larger intervals. This fact was hinted at in my discussion with the authors, and it should be made clear in the text.
- There is no evidence that QTC succeeds in estimating the threshold beta. In my discussion with the authors, I mentioned a synthetic experiment where the true value of beta is known should be included to show convergence of the estimator. The authors included a link to a Jupyter notebook, but no changes were made to the manuscript.

Overall, I think the idea behind the submission is novel and could be of interested to the community, but the current version of the manuscript is not ready for publication in TMLR: claims should be made clearer and evidence stronger in order to recommend acceptance.

***********

**Audience:**

Yes. the problem solved is important for the ML community.  All reviewers agree on this point

**Claims And Evidence:**

No.  The theory and experiment do not support the claimed advantage of the proposed method.

**Resubmission Of Major Revision:**

The authors may consider submitting a major revision at a later time.